# 3D visualization of additive occlusion and tunable full-spectrum fluorescence in calcite

David C. Green[1], Johannes Ihli[1,†], Paul D. Thornton[1], Mark A. Holden[1,2], Bartosz Marzec[1], Yi-Yeoun Kim[1], Alex N. Kulak[1], Mark A. Levenstein[1,3], Chiu Tang[4], Christophe Lynch[5,6], Stephen E.D. Webb[5], Christopher J. Tynan[5] & Fiona C. Meldrum[1]

From biomineralization to synthesis, organic additives provide an effective means of controlling crystallization processes. There is growing evidence that these additives are often occluded within the crystal lattice. This promises an elegant means of creating nanocomposites and tuning physical properties. Here we use the incorporation of sulfonated fluorescent dyes to gain new understanding of additive occlusion in calcite ($CaCO_3$), and to link morphological changes to occlusion mechanisms. We demonstrate that these additives are incorporated within specific zones, as defined by the growth conditions, and show how occlusion can govern changes in crystal shape. Fluorescence spectroscopy and lifetime imaging microscopy also show that the dyes experience unique local environments within different zones. Our strategy is then extended to simultaneously incorporate mixtures of dyes, whose fluorescence cascade creates calcite nanoparticles that fluoresce white. This offers a simple strategy for generating biocompatible and stable fluorescent nanoparticles whose output can be tuned as required.

[1] School of Chemistry, University of Leeds, Woodhouse Lane, Leeds LS2 9JT, UK. [2] School of Earth and Environment, University of Leeds, Leeds LS2 9JT, UK. [3] School of Mechanical Engineering, University of Leeds, Woodhouse Lane, Leeds LS2 9JT, UK. [4] Diamond Light Source, Harwell Science and Innovation Campus, Didcot OX11 0DE, UK. [5] Central Laser Facility, Science and Technology Facilities Council, Research Complex at Harwell, Rutherford Appleton Laboratory, Didcot OX11 0QX, UK. [6] London Centre for Nanotechnology, UCL, London WC1H 0AJ, UK. † Present address: Paul Scherrer Institute, 5232 Villingen, Switzerland. Correspondence and requests for materials should be addressed to F.C.M. (email: F.Meldrum@leeds.ac.uk).

The incorporation of guest species within host materials is an attractive route to the formation of new functional materials and promises the opportunity to tailor the properties of composites at the nanoscale level[1]. Porous materials such as zeolites, organic cages and metal organic frameworks (MOFs) provide natural candidates for hosts, where occlusion of partner molecules, or even nanoparticles, is beneficial in a variety of applications, including storage and sorting[2], catalysis[3], lighting[4] and delivery[5]. Organic macromolecules offering cavities, such as cyclodextrins, have also been widely explored for use as biosensors and drug delivery agents[6]. In all of these cases, the ability to incorporate guest species depends on a match between the size, structure and charge of the guest species and the host.

It is of course also possible to introduce guest species into non-porous crystalline hosts, forming traditional solid solutions[7,8]. Foreign ions with appropriate size and charge can be exchanged for ions of the parent lattice, and judicious selection of the dopant can create a material with new optical, magnetic and electronic properties. Interestingly, this strategy can be extended to the doping of single crystals with a wide range of species, providing a versatile method for tailoring properties. One of the best-studied systems is the biominerals, where occlusion of biomacromolecules, either individually or as aggregates can lead to superior mechanical properties[9,10]. Taking inspiration from this biogenic strategy, a range of organic and inorganic particles have been occluded within calcite ($CaCO_3$), and located within the lattice using microscopy techniques[11–16]. Again, these occlusions can be used to endow the host with new properties, such as increased hardness, magnetism and colour. Recent work has also shown that occlusion of amino acids within ZnO crystals can be used to tune the bandgap of this semiconductor[17].

A particularly elegant approach to creating single-crystal composites is through the incorporation of dye molecules, where this can generate functional materials such as tunable lasers[18,19]. One of the real strengths of this strategy, however, is that it also provides a unique method for understanding additive/crystal interactions and occlusion mechanisms. The signature colour or fluorescence of the dyes immediately reveals their location, while changes in emission spectra give information on their environments within the crystal[20,21].

Here we use the incorporation of fluorescent dyes to investigate mechanisms of organic additive occlusion within calcite. While calcite is one of the most widely studied crystals, due to its excellence as a model system and its biological, environmental and industrial importance, many questions remain concerning the mechanisms by which additives control its growth. Our study uses confocal fluorescence microscopy (CFM) to demonstrate that organic additives can occlude within calcite in specific zones, while fluorescence spectroscopy and fluorescence-lifetime imaging microscopy (FLIM) reveal the existence of different local environments within the crystals. These results are compared with dye occlusion in amorphous calcium carbonate. Having established the occlusion of individual dyes within calcite, we then extend our study to create a functional material—white fluorescent calcite—through the simultaneous incorporation of red, blue and green fluorescent dyes, which together yield a fluorescence cascade. This one-pot synthesis provides a low-cost and versatile method for generating a biocompatible, fluorescent material whose output can be tuned as required, and where the host crystal ensures greater photostability by protecting the occluded dyes from fluorescence quenchers, humidity and oxidation.

## Results

### Crystal growth studies

Calcium carbonate was precipitated in the presence of three fluorescent dyes (Supplementary Fig. 1).

A green-emitting dye HPTS (8-hydroxypyrene-1,3,6-trisulphonic acid)—termed 'GREEN' throughout for simplicity, a blue-emitting naphthalene-based dye HNDS (3-hydroxynaphthalene-2,7-disulphonic acid)—termed 'BLUE' and a newly synthesised orange-emitting perylene-based dye ($N,N'$-bis(ethyl-2-sulphonic acid)-1,6,7,12-tetrakis(phenoxy-4-sulphonic acid)perylene-3,4,9,10-tetracarboxylic acid-diimide)—termed 'RED' were selected for their high solubility in water, and for their sulphonate functionality, which was expected to provide a strong interaction with calcite[22,23]. Further investigation of the dye occlusion within calcite and the relationship between occlusion and morphology was carried out using GREEN. We then built on these data by simultaneously occluding GREEN, BLUE and RED within micron and nano-sized calcite, thereby generating calcite crystals with tunable fluorescence spectra.

### Zoning of GREEN fluorescent dye in calcite

$CaCO_3$ was precipitated in the presence of GREEN by combining equimolar solutions of $CaCl_2$, and either $Na_2CO_3$ (5 or 25 mM) or $NaHCO_3$ (3.5 mM). ACC is precipitated as the first phase under the former, but not under the latter conditions. Composite calcite crystals grown from $[Ca^{2+}] = [CO_3^{2-}] = 25$ mM with [GREEN]/$[Ca^{2+}] = 0.02$ exhibited perfect rhombohedral morphologies and showed the highest fluorescence intensity towards the upper face (that is, that furthest from the glass substrate), indicating more efficient occlusion at later growth stages (Fig. 1a–e). This was confirmed from grey value line profiles, which showed that fluorescence intensity increased from the centre of the crystal towards the faces (Fig. 1e). Crystals additionally displayed a characteristic lower intensity cross spanning opposite vertices, where the CFM image presents a slice through the centre of the crystal.

Rhombohedral crystals were also precipitated under conditions of slower crystal growth ($[Ca^{2+}] = [CO_3^{2-}] = 5$ mM and [GREEN]/$[Ca^{2+}] = 0.02$) (Fig. 1f–j), where the dye preferentially located within one half of the crystals only. This is characteristic of intra-sectoral zoning. Only a sub-volume of a single growth sector (associated with symmetry-related crystal faces) takes up the additive[18,24], where this is attributed to preferential association of an impurity with specific hillock steps[25]. Inter-sectoral zoning, in contrast, describes the uniform occlusion of dye within a particular growth sector. That soluble additives can preferentially occlude within specific zones in calcite single crystals is recognized for ions such as $Mg^{2+}$, $Mn^{2+}$, $Sr^{2+}$ (refs 26–28) and $SO_4^{2-}$ (ref. 29), where detailed investigations by atomic force microscopy (AFM) have demonstrated preferential binding of the smaller ions ($Mg^{2+}$ and $Mn^{2+}$) to the acute step edges and the larger ions ($Sr^{2+}$ and $SO_4^{2-}$) to the obtuse. Correlation between the location of the dye and the crystal morphologies (Fig. 2) demonstrates that GREEN preferentially associates with the acute over the obtuse step edges of calcite. This was corroborated by preliminary AFM studies (Supplementary Fig. 2). It is also noted that the pattern of dye occlusion seen in these crystals is not determined by their orientation on the substrate. The crystals shown in Figs 1f–j and 2 are precipitated under the same conditions but show different orientations. Nevertheless, they exhibit identical occlusion patterns.

The crystals precipitated at $[Ca^{2+}] = [HCO_3^-] = 3.5$ mM, by comparison, showed quite different morphologies and dye distributions. While the crystals formed at $[Ca^{2+}] = 5$ mM varied little in morphology or dye distribution with the initial concentration of GREEN (Supplementary Fig. 3), the crystals formed at $[Ca^{2+}] = 3.5$ mM were sensitive to the dye concentration and showed edge truncations and the emergence of new, roughened faces with increasing concentration of GREEN

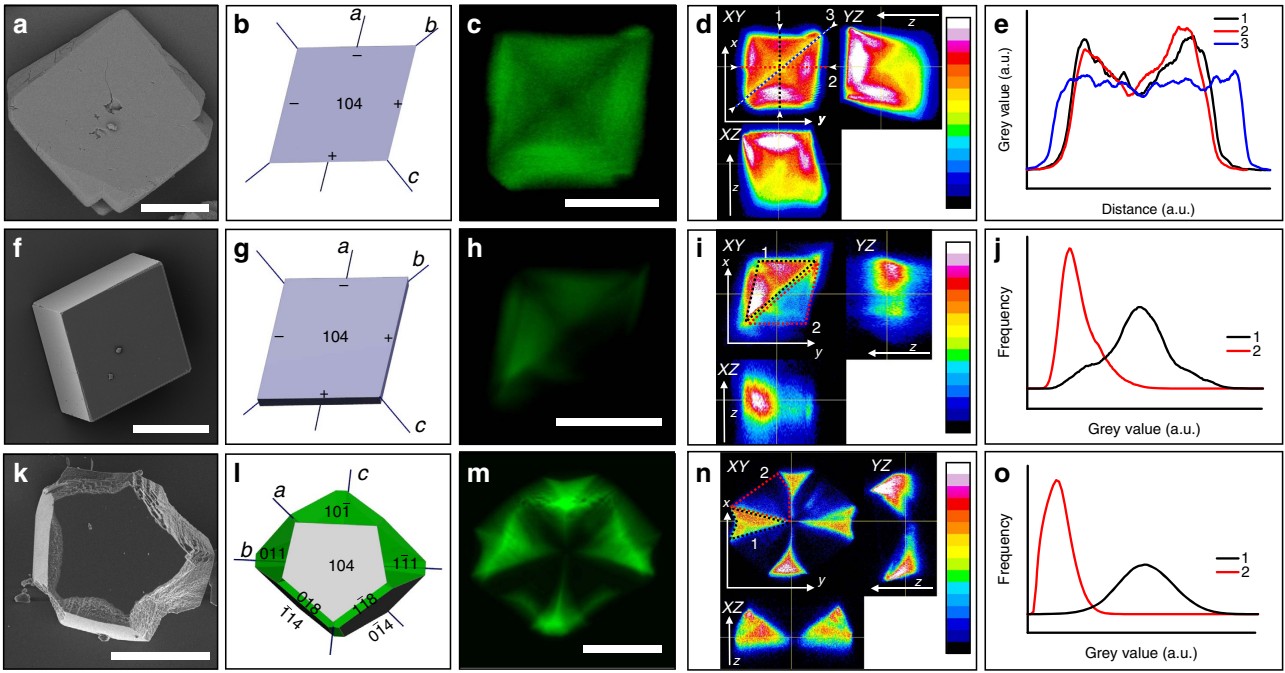

**Figure 1 | GREEN/calcite host–guest composites. (a,f,k)** Representative SEM micrographs of GREEN/calcite composite crystals precipitated under the conditions (**a–e** = $[Ca^{2+}]$ = $[CO_3^{2-}]$ = 25 mM and [GREEN] = 0.1 mM; **f–j** = $[Ca^{2+}]$ = $[CO_3^{2-}]$ = 5 mM and [GREEN] = 0.1 mM; and **k–o** = $[Ca^{2+}]$ = $[HCO_3^-]$ = 3.5 mM and [GREEN] = 0.1 mM). (**b,g,l**) Three-dimensional (3D) representation of the morphologies of the crystals imaged in **c,h,m**, with approximate faces labelled, and the growth sectors coloured in green. The + and − labels denote obtuse and acute step morphologies, respectively. (**c,h,m**) Confocal fluorescence micrographs of composite crystals grown under the conditions. (**d,i,n**) Orthogonal views (*XY*, *YZ* and *XZ*) of composites obtained from *z*-stacked confocal micrographs, showing the distribution of dye in 3D (*z*+ direction is away from the substrate, *XY* is the imaging plane) Colour scale: blue (low intensity) to red (high intensity); black, no signal; White, detector saturation. (**e,j,o**) Line profiles (**e**) and intensity histograms (**j,o**) corresponding to the lines or regions shown on the orthogonal view images in (**d,i,n**) respectively. Scale bars, 15 μm (**a**); 10 μm (**f**); 25 μm (**k**); 20 μm (**c,h,m**).

(Fig. 1k–o). CFM showed that the dye occlusion followed a Maltese cross motif whose arms expanded in width with increasing [GREEN] (Fig. 3). The fluorescence intensity was very low in the growth sectors beneath the mirror-smooth {104} faces, and was concentrated in growth sectors terminated by rough facets. Such occupancy of symmetry-related sectors is indicative of inter-sectoral zoning[24]. Modelling of the morphologies of these crystals using the programme WinXMorph suggested that the new faces were approximately parallel to {110} and {018} faces[30,31].

**Optical properties of GREEN/calcite composites**. The spectral properties of GREEN in aqueous solution were established with fluorescence spectroscopy, where broad excitation maxima were observed at $\lambda_{ex}$ = 334 and 415 nm, and an emission maximum at 508 nm was observed on excitation at 415 nm (Fig. 4a). Occlusion of GREEN in calcite crystals precipitated at $[Ca^{2+}]$ = $[CO_3^{2-}]$ = 5 mM resulted in a change in $\lambda_{em}$ from 508 to 512 nm, and additional peaks emerged at longer wavelengths with increasing levels of incorporation (Fig. 4c). Changes in excitation and emission spectra are frequently seen on the occlusion of dyes within crystals due to changes in their conformations or local environments[18]. For example, rhodamine[19], aniline[32] and pyrene-based dyes[19] occluded in $K_2SO_4$ show red shifts in the emission maxima, while blue shifts of absorption maxima and either red or blue shifts in emission maxima were observed for dyes incorporated within potassium dihydrogen phosphate[33,34]. Notably, however, in addition to a small shift in the primary emission maximum, we also observe a significant increase in the intensity of a secondary peak at 550–553 nm with increasing

levels of occluded dyes. This spectral change is often observed in more concentrated solutions of dyes, and is attributed to H-type π-stacking[35–37]. As HPTS is anionic under the reaction conditions used, such stacking could be promoted by ion-bridging by $Ca^{2+}$ ions, and the contribution of π–π stacking. Our data suggest, therefore, that both dye stacking and local environmental changes may contribute to the spectral changes observed here.

GREEN/calcite crystals were also investigated using FLIM to gain information about the local environment of the dye within the crystal lattice (Fig. 5). FLIM was conducted on GREEN/calcite crystals prepared from $[Ca^{2+}]$ = $[CO_3^{2-}]$ = 10 (Fig. 5a–d) and 2.5 mM (Fig. 5e–h), with [GREEN]/$[Ca^{2+}]$ = 0.02. Rhombohedral crystals formed under both sets of conditions, where the 10 mM samples were primarily (001) oriented (Fig. 5a). CFM micrographs revealed different patterns of dye distribution, where the crystals precipitated from 10 mM reagents exhibited more intense fluorescence at the centre of the crystal as compared with the vertices and edges, and an internal cross pattern (Fig. 5b). Those formed at 2.5 mM, in contrast, exhibit preferential location of the dye in one-half of the crystal only. In addition, the crystal shown in Fig. 5f shows that dye is concentrated beneath the small, new triangular faces formed at the crystal vertices.

The FLIM analysis also revealed changes in the fluorescence lifetime according to the location of the dye (Fig. 5c,g). While GREEN in the interior of the crystals precipitated at 10 mM had average lifetimes of $\tau$ = 4.1 ns, regions of lower intensity showed shorter lifetimes of $\tau$ = 3.2 ns (Fig. 5c,d). Crystals generated from 2.5 mM reagents yielded $\tau$ = 3.1 ns from the half-occupied interior, $\tau$ = 2.9 ns towards the surface of the crystal and significantly shorter values of $\tau$ = 2.3 ns from the small, triangular

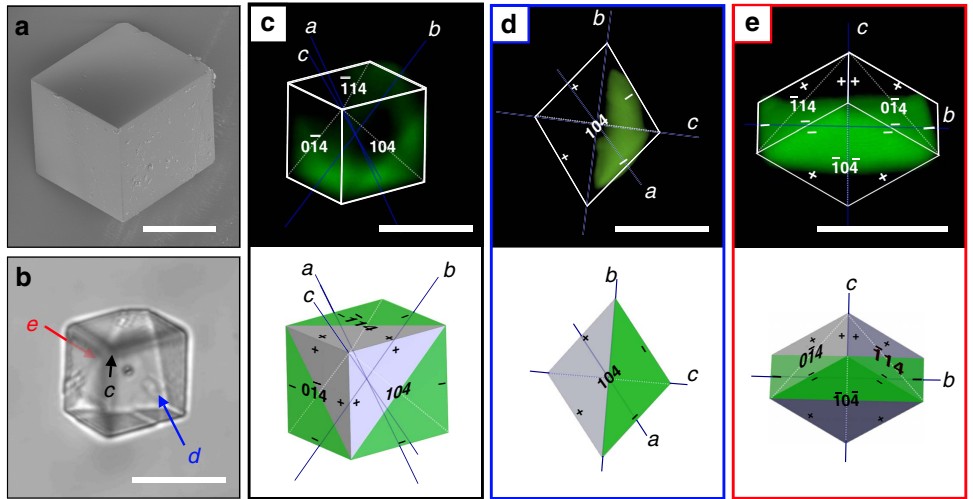

**Figure 2 | Three-dimensional distribution of GREEN in GREEN/calcite composites.** SEM (**a**) and optical (**b**) micrographs of a GREEN/calcite composite grown under conditions $[Ca^{2+}] = [CO_3^{2-}] = 5$ mM and [GREEN] = 0.1 mM oriented with crystallographic $c$ axis on a glass slide. (**c,d,e**) $z$-stacks obtained by fluorescence confocal microscopy rendered into three dimensions (3D) with accompanying model. The rendered 3D model was viewed as shown in the optical image (**c**), orthogonally to {104} face (**d**) and down the crystallographic $a$ axis (**e**). The occupied zones dominated by growth at acute steps are shaded green on modelled images, and correspond well to confocal images. Scale bars, 20 µm (**a–e**).

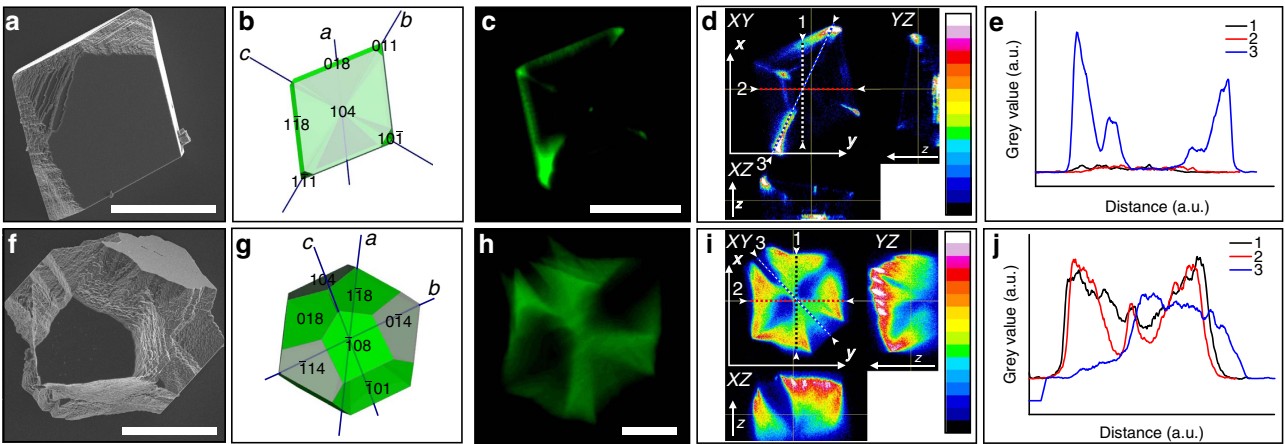

**Figure 3 | Changing GREEN distribution with changing initial [Dye].** (**a,f**) Representative SEM micrographs of GREEN/calcite composites grown from different conditions (**a–e** = $[Ca^{2+}] = [HCO_3^-] = 3.5$ mM and [GREEN] = 0.01 mM; **f–j** = $[Ca^{2+}] = [HCO_3^-] = 3.5$ mM and [GREEN] = 0.2 mM). (**b,g**) Three-dimensional (3D) representation of composites as imaged by CFM, with approximate faces labelled, and growth sectors coloured in green. (**c,h**) Confocal fluorescence micrographs of composites from the same conditions. (**d,i**) Orthogonal views images (*XY, YZ* and *XZ*) of composites obtained from $z$-stacked confocal micrographs effectively detailing the distribution of dye in 3D ($z+$ direction is away from the substrate, *XY* is the imaging plane). Colour scale: blue (low intensity) to red (high intensity); black, no signal; white, detector saturation. (**e,j**) Image analyses in the form of line profiles (**e**) and intensity histograms (**j**) corresponding to lines or regions as denoted on orthogonal view images in (**d,i**), respectively. Scale bars, 20 µm (**a,f,c,h**).

facets showing the most intense fluorescence (Fig. 5g,h). These results suggest that the incorporated dyes can occupy different environments within the lattice, where this may reflect different modes of incorporation. For comparison, dry and wet (in aqueous solution) GREEN yielded fluorescence lifetimes of $\tau = 0.6$ and 5.4 ns, respectively (Supplementary Fig. 4).

**GREEN occlusion in ACC.** Further insight into dye occlusion within $CaCO_3$ was obtained by investigating incorporation into ACC. ACC was precipitated from $[Ca^{2+}] = [CO_3^{2-}] = 25$ mM solution in the presence of GREEN. Characterisation of the ACC particles using scanning electron microscopy (SEM) and transmission electron microscopy (TEM) showed that they were 50–100 nm in size, and the polymorph was confirmed by selected-area electron diffraction, powder X-ray diffraction, Fourier transform infra-red spectroscopy and Raman spectroscopy

(Supplementary Fig. 5). GREEN/ACC composites were fluorescent under ultraviolet excitation, demonstrating the retention of GREEN within the particles (Supplementary Fig. 6). However, quantitative analysis indicated that occlusion levels (0.0014 mol%) were less than half those found in calcite (0.0034 mol%) precipitated under identical conditions ($[Ca^{2+}] = [CO_3^{2-}] = 25$ mM, [GREEN]/$[Ca^{2+}] = 0.004$; Supplementary Figs 7 and 8). No secondary emission peaks were observed by fluorescence spectroscopy (unlike in the crystalline hosts), even at high [GREEN].

The difference in fluorescence of GREEN occluded in ACC and calcite was also demonstrated by allowing a small amount of crystallization of ACC to calcite. While calcite crystals were not distinguishable in the mineral powder by optical microscopy (Fig. 6a), they were readily distinguished using CFM thanks to their higher brightness as compared with the ACC background

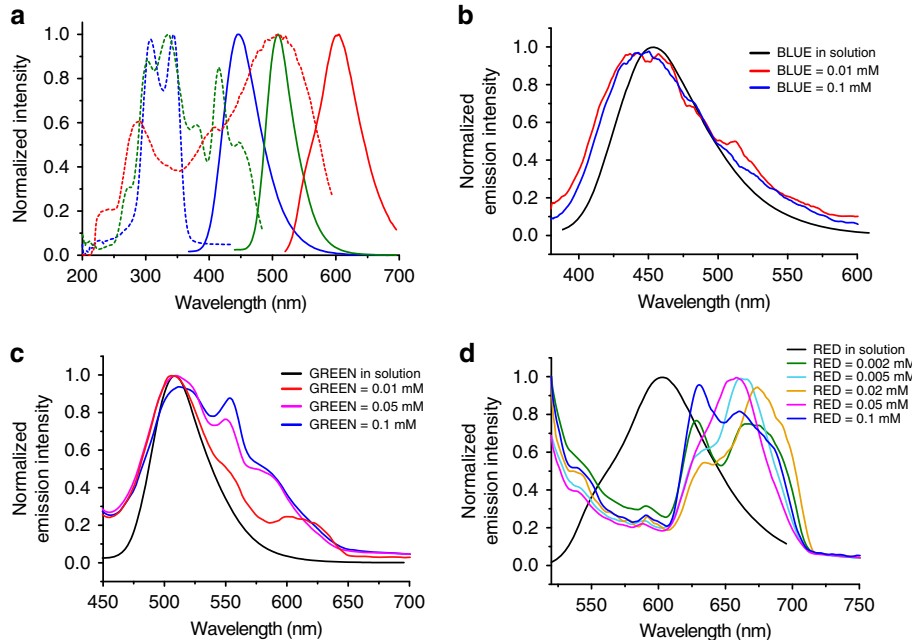

**Figure 4 | Effect of occlusion on photophysical properties of fluorescent dyes.** (**a**) Excitation (dotted) and emission (solid) spectra for aqueous solutions of BLUE (blue), GREEN (green) and RED (red). (**b**–**d**) Emission spectra of dye/calcite composites of different initial concentrations of dyes in $[Ca^{2+}] = [CO_3^{2-}] = 5\,mM$ experiments (**b**) BLUE, (**c**) GREEN and (**d**) RED. For clarity, the spectra deriving from different starting concentrations of dyes are indicated in the legends.

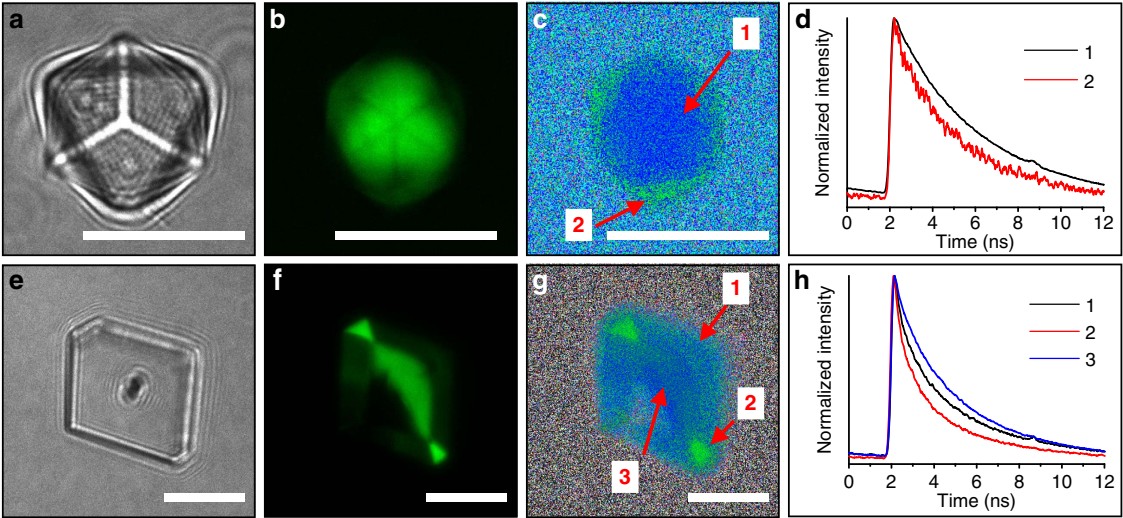

**Figure 5 | FLIM analysis of GREEN/calcite composites.** Samples prepared under (**a**–**d**) $[Ca^{2+}] = [CO_3^{2-}] = 10\,mM$ and (**e**–**h**) 2.5 mM conditions were characterized by (**a,e**) optical and (**b,f**) CFM. (**c,g**) FLIM analysis of the same plane as that in the confocal image revealed regions of differing fluorescence lifetime. (**d,h**) Global fluorescence decays obtained from the regions of interest labelled in (**c,g**) yielded fluorescence lifetimes (**d**) 1 = 4.1 ns and 2 = 3.2 ns, and (**h**) 1 = 2.9 ns, 2 = 2.3 ns and 3 = 3.1 ns. Scale bars, 10 μm (**a–c,e–g**).

(Fig. 6b). This difference was also observed on comparing the average grey values from areas attributed to amorphous and crystalline phases, where FLIM demonstrated lifetimes for dyes associated with amorphous and crystalline hosts of $\tau = 5.2$ and 3.9 ns, respectively (Fig. 6c,d).

This behaviour was attributed to the very different environments provided by calcite and ACC, where the significant levels of structural water in ACC[38] provide a more hydrated environment that may reduce the aggregation of the dyes. This hypothesis was supported by FLIM data, where the fluorescence lifetime associated with GREEN/ACC is almost identical to that of aqueous GREEN ($\tau_{ACC} = 5.2$ ns cf. $\tau_{aq} = 5.4$ ns)

(Supplementary Fig. 4). That the ACC contained significantly less GREEN than calcite crystals grown in the same concentration of dye is also interesting and perhaps at first sight counterintuitive, given that the dye could be expected to occlude more readily into the disorganised ACC structure than the crystalline calcite phase. However, dye occlusion is a kinetically driven process such that the mechanism by which occlusion occurs must be considered together with the stability of the product. Our data therefore demonstrate that binding of the dye to the calcite steps, and subsequent overgrowth is a more efficient process than co-precipitation with calcium and carbonate ions during the formation of ACC.

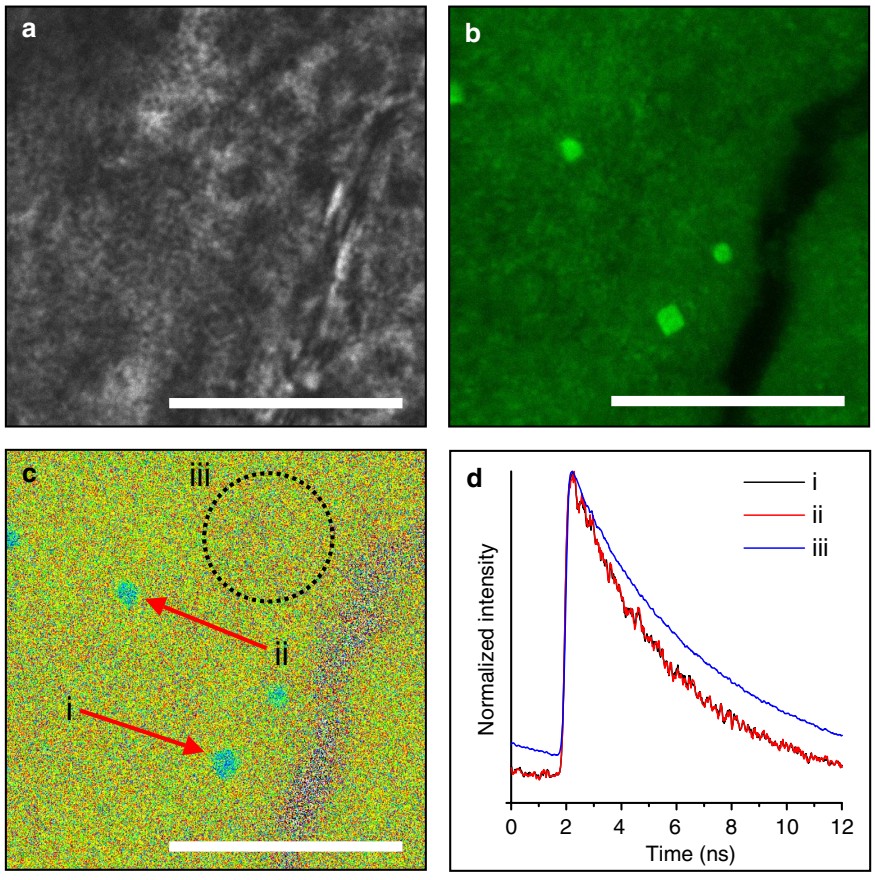

**Figure 6 | Direct comparison of local GREEN environment by FLIM. (a)** Optical micrograph of ACC after extended exposure to reaction solution. **(b)** CFM reveals calcite embedded amongst bulk ACC, with greater fluorescence intensity associated with the crystalline phase compared with the amorphous. **(c)** FLIM micrograph revealing differences in fluorescent lifetime between crystalline (i and ii) compared with amorphous (iii). **(d)** Global fluorescence decays obtained for regions of interest i (black), ii (red) and iii (blue) in **c** yielding fluorescence lifetime $\tau = 3.9$, 3.9 and 5.2 ns, respectively. Scale bars, 20 µm (**a–c**).

**Full-spectrum white fluorescent calcite**. The generality of incorporating sulphonated fluorescent dyes in calcite was further investigated by occluding BLUE and RED within calcite. $CaCO_3$ was precipitated in the presence of both dyes under conditions $[Ca^{2+}] = [CO_3^{2-}] = 5$ mM and $[Ca^{2+}] = [HCO_3^{-}] = 3.5$ mM under various $[Dye]/[Ca^{2+}]$ conditions. BLUE did not modify the typical rhombohedral morphology under either growth condition, even at higher BLUE concentrations (Supplementary Figs 9 and 10). RED caused plate-like overgrowths on single-crystal cores at $[Ca^{2+}] = [CO_3^{2-}] = 5$ mM and $[RED]/[Ca^{2+}] = 0.02$ (Supplementary Fig. 11) while rhombohedral morphologies were observed at $[RED]/[Ca^{2+}] = 0.001$. Very low [RED] were also used for $[Ca^{2+}] = [HCO_3^{-}] = 3.5$ mM ($[RED]/[Ca^{2+}] = 0.0003$ and 0.0006), and generated some new, small {001} faces, where this morphology is characteristic of obtuse step blocking (Supplementary Fig. 10)[39].

The optical properties of BLUE and RED within calcite were also investigated. Broad excitation maxima were observed in aqueous solution at $\lambda_{ex} = 308$ and 324 nm for BLUE and 506 nm for RED, while blue and orange emission maxima at 458 and 603 nm were observed on excitation of BLUE and RED at 308 and 506 nm, respectively (Fig. 4a). In dye/calcite composites formed at $[Ca^{2+}] = [CO_3^{2-}] = 5$ mM, little change was observed with BLUE (Fig. 4b) while RED showed significant red shifts, where the principle emission peak observed in solution was replaced by lower energy peaks at 628–633 and 659–673 nm. As with GREEN, this was attributed to stacking of the dye molecules (Fig. 4d).

To use dye incorporation to tune the fluorescence signal of the host calcite crystals it was necessary to quantify the relative occlusion efficiencies of the individual dyes. This was achieved by quantifying the dye liberated from solubilized crystals. All dyes showed an approximately linear increase in occlusion with the initial $[Dye]/[Ca^{2+}]$, where a similar trend has been observed with the occlusion of aspartic acid and glycine in calcite (Supplementary Fig. 7)[40]. RED incorporated to the highest extent, followed by BLUE and GREEN; under conditions of $[Ca^{2+}] = [CO_3^{2-}] = 5$ mM and $[Dye]/[Ca^{2+}] = 0.004$, RED, BLUE and GREEN are occluded to levels of $\approx 0.05$, 0.02 and 0.002 mol%, respectively. This suggests that incorporation is related to the number and distribution of the sulphonate groups rather than the molecular weight of the dye. While these levels of incorporation were too low to give significant lattice parameter distortion or line broadening from high-resolution synchrotron powder X-ray diffraction analysis (Supplementary Fig. 12), the composite crystals exhibited visible fluorescence under ultraviolet excitation (365 nm) (Supplementary Fig. 13).

Calcite was then precipitated in the presence of mixtures of the three dyes, where their simultaneous incorporation was anticipated to yield full-spectrum white fluorescence on ultraviolet excitation (Fig. 7a). Owing to the different $\lambda_{ex}$ maxima of the individual dyes, white fluorescence was expected due to a fluorescence cascade; blue-emitted light from BLUE excites GREEN, which in turn emits green light that excites RED (Fig. 7b). Reaction conditions were selected based on the ratio of

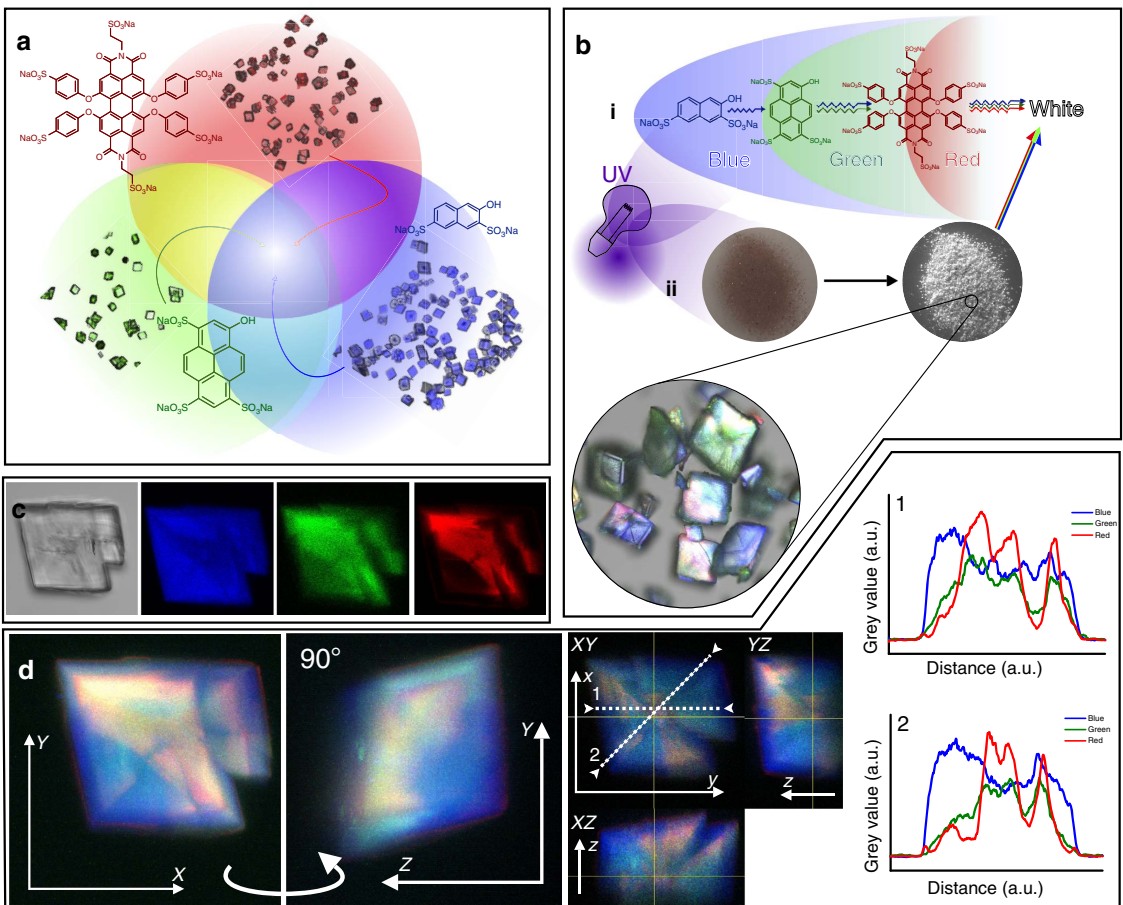

**Figure 7 | White calcite. (a)** A mixture of blue, green and red light emitted from crystals containing BLUE, GREEN and RED, respectively, would yield a full-spectrum white light. **(b)** The fluorescence cascade concept (i), where ultraviolet radiation excites a blue fluorescent dye, which in turn excites a green fluorescent dye, which finally excites a red fluorescent dye. With appropriate proportions of each dye, an equal intensity of blue, green and red light will be emitted, yielding white light. **(c)** Optical and confocal fluorescence micrographs of a sample calcite/dye composite containing all three dyes. **(d)** z-stack of confocal fluorescence micrographs rendered in three dimensions at obtained and rotated through 90° along y axis; Orthogonal views plot of calcite containing all three dyes and line profiles of each colour as denoted.

the dyes in the reaction solution required to obtain suitable incorporation levels of each dye at $[Ca^{2+}] = [CO_3^{2-}] = 5\,mM$. To yield a white cascade the dyes must be present at levels where a suitable proportion of emission photons, particularly from BLUE and GREEN, is not absorbed. They therefore contribute their respective colours to the net fluorescence. A range of conditions were explored (Supplementary Table 1), and dye mixture 5 yielded white fluorescent calcite as confirmed by optical characterization (Supplementary Fig. 13).

The distribution of the three occluded dyes within the crystals was determined using CFM (Fig. 7c). BLUE and RED showed comparable patterns of localization as the single-dye studies (Fig. 7d), while GREEN showed preferential localization in regions dominated by RED, yielding net yellow fluorescence. These effects can be attributed to favourable pyrene–naphthalene and pyrene–perylene interactions, which result in mixed π-stacking, and comparatively poor naphthalene–perylene interactions[41]. Despite some inhomogeneities in the distributions of the fluorescent dyes, however, the close proximity of the dyes in the crystal and the formation of a fluorescence cascade ensures bulk white fluorescence is achieved.

**Fluorescent calcite nanoparticles.** Finally, our concept of fluorescent dye occlusion was extended to calcite nanoparticles. These were formed via a modified carbonation method

(Supplementary Figs 14 and 15)[42], and TEM and powder X-ray diffraction analysis revealed 55 nm calcite particles (Fig. 8 and Supplementary Fig. 16). The synthesis was performed in the presence of individual or mixtures of dyes, and neither the morphologies nor particle sizes were significantly affected by the presence of dye. The calcite nanoparticles exhibited bright fluorescence from both dried nanoparticles (Supplementary Fig. 17) and ethanolic suspensions (Fig. 8c–g) on excitation with ultraviolet light, where the emitted light was dependent on the dye (or mixture of dyes) present during nanoparticle growth. Quantification of the amounts of fluorescent dye occluded revealed an identical trend to that seen for micron-scale calcite (that is, RED > BLUE > GREEN) (Supplementary Table 2), and no significant differences in the spectroscopic data were observed as compared with the micron-scale calcite crystals. The fluorescence lifetime of GREEN in nanoparticulate calcite was $\tau = 3.0\,ns$, which is comparable to values obtained for internal regions of calcite single crystals grown from $[Ca^{2+}] = [CO_3^{2-}] = 2.5\,mM$.

**Discussion**

There is a huge amount of literature on the influence of organic additives on the precipitation of $CaCO_3$, where these have focused on the effect on crystal polymorph and morphology[43]. Owing to the challenge of quantifying the amounts of organic molecules

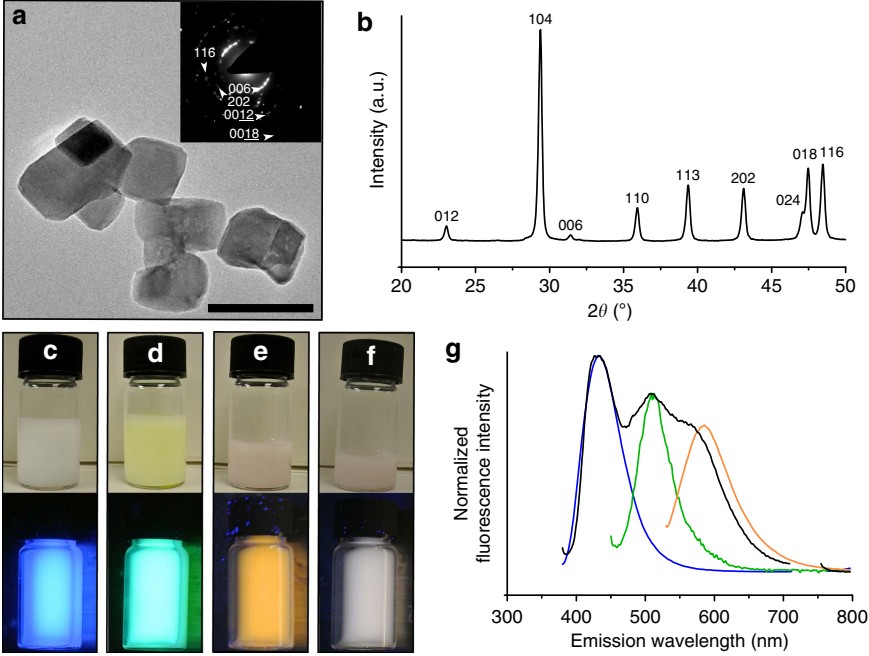

**Figure 8 | Fluorescent dye/calcite composite nanoparticles.** (**a**) Representative TEM micrograph of calcite nanoparticles occluding GREEN and selected-area electron diffraction pattern (inset). (**b**) Powder X-ray diffraction analysis of calcite nanoparticles demonstrating wide-line broadening associated with small crystalline domain size ($\tau = 53$ nm). (**c**–**f**) photographs of ethanolic suspensions of fluorescent calcite nanoparticles containing (**c**) BLUE, (**d**) GREEN, (**e**) RED and (**f**) dye mixture under normal light (top) and ultraviolet light (365 nm, bottom). (**g**) Emission spectra of ethanolic suspensions of calcite nanoparticles occluding BLUE (blue, $\lambda_{ex} = 360$ nm), GREEN (green, $\lambda_{ex} = 430$ nm), RED (orange, $\lambda_{ex} = 512$ nm) and dye mixture (black, $\lambda_{ex} = 360$ nm; section at $\sim 700$–760 nm removed due to signal from excitation light at $2\lambda_{ex}$). Scale bar, 100 nm (a).

present within a carbonate-based crystal, much less is known about whether association of the additives with the growing crystals is accompanied by their incorporation into the crystal lattice. It is well recognized that proteins are occluded within calcite single-crystal biominerals, where these are quantified by gravimetric analysis after dissolution of the mineral[44]. Our study provides new insight into the occlusion of small organic molecules (here fluorescent dyes) within calcite single crystals, demonstrating the existence of zoning and the relationship between occlusion and changes in crystal morphologies. It is noted that proteins tagged with fluorescent dyes have been occluded within polycrystalline and single-crystal calcite, but zoning was not observed[45]. The dye Congo Red has also been reported to occlude within calcite[46], and a recent AFM study has showed that this dye stabilizes the polar step edges along the [010] direction of calcite (10.4)[47].

Traditionally, the emergence of new crystal faces on calcite was attributed to the additives preferentially adsorbing to, and stabilizing these faces[44,48–50]. *In situ* AFM studies modified this view and demonstrated that the morphological changes caused by species such as aspartic acid arise due to the additives binding to the {104} step edges[51]. Changes in the shape and separation of these step edges then translate into changes in the macroscopic morphology. It is now also recognized that the growth of low-solubility crystals such as calcite is limited by the availability of kink sites, such that once a stabilized growth unit has formed on a step edge, it can readily propagate[52–55]. Impurities can therefore interfere with this process by binding to kink sites as they form, or during their propagation. Binding can be considered irreversible (when the detachment rate is extremely slow as compared with the rate of step propagation) or reversible, where the residence time of the impurity at a kink site as compared with the propagation rate of the steps will determine how efficiently the dye is occluded.

Both the incorporation of GREEN and its effects on calcite morphology vary systematically with the supersaturation of the crystal growth solution. While uniform occlusion is observed at $[Ca^{2+}] = [CO_3^{2-}] = 25$ mM and $[GREEN] = 0.1$ mM (Saturation Index (SI) = 3.5), intra-sectoral zoning was observed at the lower supersaturation conditions present at $[Ca^{2+}] = [CO_3^{2-}] = 5$ mM and $[GREEN] = 0.1$ mM (SI = 2.7) (Fig. 1). This demonstrates that the dye binds to both the acute and obtuse steps, but that binding is stronger to the acute sites. This transition also shows that the binding of the dye to the kink sites is reversible, and that the average residence time of a dye molecule at an acute kink site is longer than at an obtuse step; this is additionally supported by the low level of incorporation of GREEN. Notably, rhombohedral crystals formed under both sets of conditions, which demonstrates that a change in crystal morphology cannot be used as a signature for additive occlusion within the crystal lattice.

At the higher supersaturation, there is an increased probability of forming new kink sites, which creates more opportunities for binding and incorporating impurities[52]. While these impurities will necessarily block the propagation of the sites to which they are adsorbed, adjacent active (unblocked) kink sites can continue to grow until they permanently entrap the impurity. As kink sites are on average closer together at higher supersaturations, occlusion will be more effective. Faster growth also gives rise to occlusion at both acute and obtuse steps. If growth is sufficiently rapid as compared with the rate of detachment from the obtuse step, little difference will be seen in the occlusion at the acute and obtuse steps. As the supersaturation is reduced, the average density of kink sites and the rate of completion of step edges are reduced, and the average separation of the kink sites is increased. Under these conditions the shorter residence time of the dye at the obtuse step becomes important such that only dyes bound to acute steps are resident long enough to become occluded.

More complex inter-sectoral zoning in the form of Maltese-cross-type patterns was observed on further reduction of the supersaturation to conditions of $[Ca^{2+}] = [HCO_3^-] = 3.5\,mM$ and $[GREEN] = 0.1\,mM$ (SI = 0.9). This was accompanied by changes in the crystal morphologies such that they exhibited both mirror-smooth {104} faces and highly roughened faces approximately parallel to {011} and {018} surfaces. Dye occlusion only occurred in zones associated with the rough crystal faces, and virtually no dye was located in zones associated with the smooth {104} faces. Increase of the dye concentration (under otherwise identical solution conditions) resulted in an increase in the relative area of the rough surfaces, and a corresponding increase in the volume of the sectors in which dye was localized, while reduction in the dye concentration to $[GREEN] = 0.01\,mM$ gave rhombohedra that only displayed slight dye occlusion at growth sector boundaries (Fig. 3).

Under these conditions of low supersaturation, the average number of kink sites on the {104} planes is small, such that there are few opportunities for dye incorporation within these faces. However, the dye molecules continue to occlude at the boundaries between different growth sectors. The intersections between zones (both inter-sectoral and intra-sectoral) typically correspond to sites of increased lattice strain that can preferentially trap impurities[56,57]. Our data also show that higher levels of occlusion at the boundaries between growth sectors correlate with changes in the crystal morphologies. Enhanced occlusion at these sites disrupts the normal translation of the steps on the {104} faces, thereby generating new, striated faces. As these are stabilized by the adsorbed/occluded dye molecules, their growth is slower than the smooth {104} faces such that their relative sizes increases during crystal growth. It is also noted that calcite crystal morphologies are the result of a complex interplay between multiple variables. The ratio of $Ca^{2+}$ to $CO_3^{2-}$ in the growth solution, the pH and the ionic strength can all affect the growth rate of calcite at fixed values of the supersaturation and that the acute and obtuse steps can grow at different rates according to the solution conditions[56,58,59].

Our demonstration that the fluorescent dyes can occlude within the calcite crystals by different mechanisms—binding to kink sites (leading to occlusion within zones associated with smooth {104} faces) or at the boundaries between growth sectors (leading to occlusion within zones associated with rough faces)—is also supported by the fluorescent lifetime measurements. Occlusion associated with the {104} faces was characterized by significantly longer lifetimes ($\tau = 4.1$–$3.2\,ns$) than occlusion in the rough faces ($\tau = 2.3\,ns$). These values can also be compared with the same dye in solution ($\tau_{aq} = 5.4\,ns$) and in a dry state ($\tau_{dry} = 0.6\,ns$). This provides a clear demonstration that dyes occluded under high and low rates of growth occupy different environments within the crystal.

To-date, information about the location of organic additives within calcite has predominantly come from single-crystal[60–62] and powder[63] X-ray diffraction studies of biominerals and synthetic crystals precipitated in the presence of additives. Anisotropic lattice distortions are observed, which provide insight into the orientation of the additives within the crystals. It is noted, however, that calcite is elastically anisotropic and more flexible along the $c$ axis than the $a$ axis[40]. Therefore, greater distortion along the $c$ axis does not necessarily mean that the additive is located on planes perpendicular to the $c$ axis. Single-crystal X-ray diffraction data have also been correlated with observed morphological changes to link additive adsorption to calcite to occlusion[60–62]. However, the demonstration that additives bind to {104} step edges has superseded this picture[51]. Indeed, rough crystal faces on calcite—regardless of the crystallographic assignment—comprise {104} faces on a microscopic scale (Supplementary Fig. 18). Additives may also reorient during occlusion, so their ultimate position within the lattice does not necessarily reflect their binding mode to the crystal surface. With the ability to visualize the additives within the calcite crystals (X-ray diffraction will necessarily record an average), our experiments provide unique insight into the mechanism of additive occlusion, and show how this can translate into morphological changes.

Adding to this discussion, it is also valuable to compare the mechanisms that give rise to zoning in calcite with those that operate for another common biomineral, calcium oxalate, which readily exhibits zoning effects[64–66]. Unlike calcite—which only displays smooth {104} faces—both calcium oxalate monohydrate and dihydrate exhibit a number of crystallographically distinct smooth faces. Significant differences in additive binding energies to these faces have been reported, which will facilitate inter-sectoral zoning[67,68]. A key feature of additive binding to calcite is acute/obtuse step selectivity, which drives intra-sectorial zoning. Studies of peptide binding to the different step edges present on given calcium oxalate faces suggest that the energy differences are much smaller[67,68] than those estimated for aspartic acid binding to the acute and obtuse steps on calcite[51]. This makes intra-sectorial zoning less likely for calcium oxalate than calcium carbonate.

Having established that red-, blue- and green-emitting dyes in calcite can be individually incorporated within calcite crystals, we extended this approach to the simultaneous occlusion of all three dyes. While crystals exhibiting multiple luminescence signals have been generated previously due to the sector-dependent emission of certain dyes[21], our approach enables the fluorescence signal from the crystals to be precisely tuned by simply altering the ratio of dyes in the initial solution. This strategy was also used to create calcite nanoparticles with tunable fluorescence, where it is noted that dye occlusion in calcite was more efficient than in ACC. These nanoparticles provide some clear advantages over semiconductor quantum dots in applications such as *in vivo* imaging, where they are biocompatible and degradable, and the spectral properties are independent of the surface chemistry and particle size. Calcite also has advantages over silica as a host material in that degradation products are water-soluble and easily ejected from organism. Silica, in contrast, yields water-insoluble degradation products that may exhibit cytotoxicity on accumulation[69].

This work provides new insight into the mechanisms by which organic additives occlude within calcite, and demonstrates that in common with small ions such as $Mg^{2+}$ and $Mn^{2+}$, organic molecules can concentrate within specific zones. This effect was dependent on the growth conditions, such that uniform occlusion was observed at high supersaturations, while intra-sectoral zoning occurred at lower supersaturations. The latter shows that despite the size and rigid conformation of the dye molecules, occlusion under these solution conditions is governed by preferential adsorption to acute step edges. Our experiments also provide a new understanding of the relationship between additive occlusion and the morphological development of calcite, and show that dye occlusion within the growth sectors associated with the {104} faces can occur without any shape change. Significant changes in crystal morphologies were observed at low supersaturations and high $[Ca^{2+}]/[CO_3^{2-}]$ ratios, however, where this was associated with a change in the incorporation mechanism; occlusion occurred in the growth sectors associated with new, rough crystal faces. These results highlight the complexity of additive/crystal interactions, and demonstrate that significant care must be taken when using indirect techniques to assess additive occlusion mechanisms. Finally, we extended our strategy to occlude multiple dyes within calcite crystals, where this enabled us to

generate biocompatible nanoparticles with tunable fluorescence. These fluorescent nanocomposites provide interesting potential host–guest materials for applications such as optical brighteners, pigments and bioimaging agents, where stabilized fluorophores are required.

## Methods

**Materials.** HNDS (BLUE), HPTS (GREEN), ethanol, sodium hypochlorite solution (10–15%), calcium chloride dihydrate, anhydrous sodium carbonate, anhydrous sodium bicarbonate, taurine, phenol, potassium carbonate, anhydrous pyridine, anhydrous $N,N$-dimethylformamide and ethylenediamine tetraacetic acid disodium salt ($Na_2.EDTA$) were used as purchased from Sigma-Aldrich. Sulphuric acid was used as purchased from Fluka. 1,6,7,12-tetrachloroperylene-3,4,9, 10-tetracarboxylic acid dianhydride was used as purchased from Santa Cruz Biotechnology. Deionized (DI) water was used as obtained from in-house Millipore Q system (<2 p.p.m. total organic content (TOC) and 18.2 M$\Omega$).

**Stock solutions.** Aqueous solutions of $CaCl_2.2H_2O$ (7.3505 g in 250 ml DI water for 200 mM), $Na_2CO_3$ (5.2994 g in 250 ml DI water for 200 mM) and $NaHCO_3$ (4.2003 g in 250 ml DI water for 200 mM) were prepared in 250 ml volumetric flasks and stored in glass bottles. A unit of 100 mM aqueous $Na_2CO_3$ or $CaCl_2$ solution was prepared by adding 10 ml 200 mM aqueous stock solution to 10 ml DI water, yielding 20 ml of final solution. Dye solutions were prepared by the addition of 20 ml DI water to solid dye (0.0139 g BLUE for 2 mM; 0.021 g GREEN for 2 mM; and 0.0052 g RED for 0.2 mM solution) in a glass vial. All solutions were filtered through a syringe-driven polycarbonate filter (0.22 µm) before use.

**Buffers.** An aqueous solution of $Na_2.EDTA$ dihydrate (($Na_2.EDTA.2H_2O$); 9.306 g in 250 ml DI water for 100 mM) was prepared in a 250 ml volumetric flask and stored in a glass bottle. An aqueous solution of potassium borate (pH 10.4) buffer (1 M) was prepared in a 500 ml glass beaker under constant stirring. The correct mass of boric acid (30.915 g for 500 ml 1 M solution) was added to ~300 ml DI water. Under constant monitoring of the pH, potassium hydroxide pellets were carefully added until all potassium hydroxide and boric acid had dissolved and the pH was 10.4. The solution was then decanted into a 500 ml volumetric flask and the volume made up to 500 ml with DI water before being stored in a clean glass bottle.

**Calcite/dye composite preparation.** Calcite growth studies were divided into three groups: $[Ca^{2+}] = [CO_3^{2-}] = 25$ mM; $[Ca^{2+}] = [CO_3^{2-}] = 5$ mM; and $[Ca^{2+}] = [HCO_3^-] = 3.5$ mM. Calcite single crystals were grown in beakers by direct mixing. Beakers were charged with different final volumes of reaction liquor to obtain sufficient mass of calcite/dye composites for analysis (at least 100 mg). To achieve this, total volumes of 100, 500 and 800 ml were used for $[Ca^{2+}] = 25$, 5 and 3.5 mM, respectively. Each beaker had two clean glass substrates for microscopic analysis. All experiments were prepared with dilutions of aqueous stock solutions ($[CaCl_2] = 200$ mM, $[Na_2CO_3] = 200$ mM, $[NaHCO_3] = 200$ mM, [BLUE] (HNDS) = 2 mM, [GREEN] (HPTS) = 2 mM and [RED] = 0.2 mM). Where concentrations of dye higher than those present in the stock, the solid dye was added to the beaker before addition of DI water.

For example, for $[Ca^{2+}] = [CO_3^{2-}] = 5$ mM and [GREEN] = 0.1 mM ([GREEN]/$[Ca^{2+}]$ = 0.02), 12.5 ml $CaCl_2$ stock, 25 ml GREEN stock and 450 ml DI water were mixed in a clean 1 L beaker. The solution was stirred with a clean glass rod to ensure full mixing was achieved. To initiate the reaction, 12.5 ml $Na_2CO_3$ stock was added under stirring. The beaker was covered with Parafilm and left for 3 days. The crystallization liquor was removed. Extracted glass substrates holding calcite crystals were bleached by submersion in 3.33–5% NaOCl solution, followed by rinsing with DI water and ethanol. Crystals on the surface of the beaker were then rinsed with DI water to remove remaining crystallization liquor. Crystals were scraped off the surface of the beaker in the presence of 5 ml EtOH with a spatula. When all product crystals are removed from beaker surfaces, the ethanol/crystal suspension was filtered out using Millipore vacuum filtration system (0.45 µm polycarbonate membrane). Dried crystals were bleached in 1 ml 10–15% NaOCl solution over for 24 h in centrifuge tubes. Bleached crystals were then collected by centrifugation (10 min, 8,000$g$), rinsed in DI water twice, then ethanol twice and allowed to dry. The procedure was followed for all experiments, where volumes of stock solutions and water were altered as appropriate for each experiment. In $[Ca^{2+}] = [HCO_3^-] = 3.5$ mM with equimolar $NaHCO_3$, no immediate precipitation was observed and crystallization was allowed to occur for 7 days.

**Sample preparation for quantitative studies.** Approximately 10 mg bleached, dry calcite/dye composites were precisely measured and dissolved in 3 ml 100 mM aqueous $Na_2.EDTA$ solution. The precise known mass is important for calculating mol% or wt% values of dye. A volume of 100 µl of each dissolved composite solution in $Na_2.EDTA$ was added to individual wells of a 96-well plate with 100 µl 1 M potassium borate buffer (pH 10.4) was added to each well to control pH. The concentration of previous incorporated dye was determined by fluorescence

spectroscopy with a well plate reader (Perkin Elmer EnVision 2103) and a calibration curve of known concentration of each dye.

**White calcite preparation.** White calcite samples were precipitated from $[Ca^{2+}] = [CO_3^{2-}] = 5$ mM experiments. They were performed as per other experiments, except instead of the addition of a single-dye solution, various mixtures of dyes were prepared (Supplementary Table 1)

**Sample preparation FLIM.** Samples for FLIM were prepared by direct mixing on clean glass substrates (microscope glass coverslips washed in piranha solution). All samples were prepared from 350 µl final volume of crystallization liquor. This optimized the yield and size (10–20 µm) of crystals grown directly on the substrate. Carbonate-based sample preparation was favoured due to the low signals obtained from bicarbonate-prepared samples. Different conditions ($[Ca^{2+}] = [CO_3^{2-}]$ = 2.5, 5 and 10 mM) with equimolar $Na_2CO_3$ were used, with [GREEN]/$[Ca^{2+}]$ = 0.0002, 0.002 and 0.02 for each $[Ca^{2+}]$. Crystallization was undertaken from a 1:1 mixture of a $CaCl_2$/GREEN solution and a $Na_2CO_3$ solution. These stocks were prepared immediately before the reaction took place.

For example, in the experiment $[Ca^{2+}] = [CO_3^{2-}] = 5$ mM and [GREEN]/$[Ca^{2+}]$ = 0.02, one vial was charged with 20 µl $CaCl_2$ stock (100 mM), 20 µl GREEN stock (2 mM) and 160 µl DI water, while a second was charged with 20 µl $Na_2CO_3$ stock (100 mM) and 180 µl DI water to yield two vials of total 200 µl volume in each. Next, 175 µl of $CaCl_2$/HPTS solution was carefully added by pipette to a glass substrate, followed by 175 µl of $Na_2CO_3$ solution to initiate the reaction. To yield a total of 350 µl, 1:1 volumes were used to ensure sufficient mixing on reaction initiation since nominal agitation methods (stirring, vortexing and so on) were not possible in this configuration. Crystallization occurred over 16 h in a closed Petri dish to avoid excessive evaporation before washing with water and ethanol.

**Calcite nanoparticle synthesis.** Calcite nanoparticles were generated using a carbonation method described previously[38]. A volume of 50 ml DI water was degassed/decarbonated by refluxing at 80 °C in a three-necked round bottom flask under $N_2$ flow for 6 h (Supplementary Fig. 15a). A unit of 0.44 g CaO, formed by calcination of pure $CaCO_3$ at 900 °C for 8 h, was added. Stirring, heating and $N_2$ bubbling was continued for 15 min to facilitate $Ca(OH)_2$ formation before ageing the solution in a closed vessel at room temperature for 16 h (Supplementary Fig. 15b). Enough mass of dye was then added to yield either 1 mM GREEN, 0.02 mM BLUE or 0.005 mM RED (or a mixture of all three in the same ratio as for white calcite described above) and allowed to mix with aged $Ca(OH)_2$ for 10 min until fully dissolved. Carbonation was then undertaken by bubbling a 3:1 mixture $N_2:CO_2$ through the solution at an overall flow rate of 1 l min$^{-1}$ under stirring at room temperature (Supplementary Fig. 15c). Reaction profile was monitored with pH and occurred in two stages: (1) very slow decrease in pH from initial pH 12.7–12.8 to 12.5 as $CO_2$ dissolved and buffered by $OH^-$ in solution; and (2) rapid decrease from pH 12.5 to 7 as $CO_2$ dissolved but all $Ca(OH)_2$ transformed to $CaCO_3$ (Supplementary Fig. 16). Reaction was stopped when pH = 7 by removing gas supply and closing the system. Dry, clean nanoparticles were obtained by centrifugation and twice washing with ethanol, before suspension in sodium hypochlorite (5%) solution for 2 h and further centrifugation and drying steps.

**ACC synthesis.** ACC was generated by direct mixing of 200 mM $CaCl_2$ and $Na_2CO_3$ solutions in the presence of aqueous dye (GREEN) solution of various concentration to yield a final $[Ca^{2+}] = [CO_3^{2-}] = 25$ mM. For example, for [GREEN]/$[Ca^{2+}]$ = 0.04 sample, a vial was charged with 2.5 ml 200 mM $CaCl_2$ solution, 10 ml 1 mM GREEN solution and 5 ml DI water. To initiate the reaction, 2.5 ml of 200 mM $Na_2CO_3$ solution was added, after which the vial was immediately sealed and vigorously shaken for 5 s before filtering through 0.45 µm pore polycarbonate membrane using a vacuum-driven filtration system (Millipore). Removal of water and subsequent drying with ethanol effectively quenched the reaction, preventing crystallization. ACC samples, which were subjected to partial crystallization, were prepared by the same methodology except vigorous shaking occurred for 30 s instead of 5 s. Samples were dried by vacuum and stored on desiccant and removed only immediately before analysis. The polymorph was confirmed by selected-area electron diffraction, powder X-ray diffraction (which just showed two broad peaks at $2\theta \approx 32°$ and $46°$), Fourier transform infra-red spectroscopy (which showed a peak at 3,200 cm$^{-1}$ from water and the absence of a $v_4$ peak at 714 cm$^{-1}$) and Raman spectroscopy (a single broad peak is seen at 1,085 cm$^{-1}$) (Supplementary Fig. 11).

**RED dye synthesis.** The synthetic protocol for red fluorescent sulfonated dye $N,N'$-bis(ethyl-2-sulphonic acid)-1,6,7,12-tetrakis(phenoxy-4-sulphonic acid) perylene-3,4,9,10-tetracarboxylic acid-diimide was derived from a published report (Supplementary Fig. 19)[70].

**1**: 0.5 g (0.94 mmol) 1,6,7,12-Tetrachloroperylene-3,4,9,10-tetracarboxylic acid dianhydride and 0.26 g (2.06 mmol) 2-aminoethane sulphonic acid (taurine) are added to 5 ml anhydrous pyridine and heated to 80 °C under reflux and vigorous stirring for 15 h. A dark red precipitated was collected by filtration and washed

repeatedly with ethyl acetate. Liquid chromatography/mass spectroscopy (LC/MS) (*m/z*): [$Na_2M^+$] calculated for $Na_2C_{52}H_{34}N_2O_{14}S_2$: 785.85, found 786 (dissolved in NaOH).

**2**: 0.25 g (0.335 mmol) **1**, 0.4 g (2.89 mmol) anhydrous $K_2CO_3$ and 0.26 g (2.8 mmol) phenol are added to 10 ml anhydrous *N,N*-dimethylformamide and heated to 110 °C under reflux and stirring for 15 h. After cooling, 50 ml 1 M aqueous HCl, cooled to 4 °C is added to yield a dark red suspension, which is isolated by filtration and washed repeatedly with ethyl acetate. LC/MS (*m/z*): [$M^+$] calculated for $C_{52}H_{34}N_2O_{14}S_2$: 974.15, found 977.4.

**3**: 0.25 g (0.256 mmol) **2** is added to 2 ml concentrated sulphuric acid under stirring and heating at 25 °C for 24 h. A volume of 20 ml DI water is added before dialysis against DI water for 2 days to yield aqueous suspension of **3** in DI water. Solid product is obtained via freeze drying. LC/MS (*m/z*): [$M^+$] calculated for $C_{52}H_{34}N_2O_{26}S_6$: 1293.97, found 1296.9.

**Confocal fluorescence microscopy.** CFM was conducted using a Zeiss LSM510 Upright Confocal Microscope of samples grown directly on clean glass substrates, under oil immersion where required. Low-magnification images were obtained to demonstrate dye distribution across the population of composites, and higher-magnification *z*-stacks were performed on at least three individual crystals per sample. Lasers and filters were selected based on their suitability to the excitation and emission maxima of the dye. Intensities of different lasers were required due to different dye quantum yields and overall final mol% in composites. Image rendering and analysis was conducted in ImageJ.

**Scanning electron microscopy.** SEM was conducted using an FEI NanoSEM Nova 450 from samples grown directly on clean glass substrates. Samples were mounted on aluminium stubs with double-sided Cu tape. All samples were coated with 2 nm Ir conductive layer before analysis.

**Transmission electron microscopy.** TEM was conducted using an FEI Tecnai TF20 FEG-TEM fitted with an Oxford Instruments INCA 350 EDX system/80 mm X-Max SDD detector and a Gatan Orius SC600A CCD camera operating at 200 kV. Samples were loaded onto carbon-coated Cu grids.

**Fluorescence spectroscopy.** Fluorescence spectroscopy was conducted using a Perkin Elmer LS 55 operated with WinLab software. Suspensions or solutions were analysed in a quartz cuvette, whereas dry samples were mounted on grease-coated glass slides so that they could be situated directly under the incident beam, and that they remained in place when the sample was tilted to optimize the emissive signal.

**Quantitative analysis.** Quantitative analysis was conducted of EDTA-solubilized samples of known mass against a calibration curve of known [Dye] in 96-well plates (Greiner Black μClear) to facilitate mol% dye versus $CaCO_3$ calculations. Plates were analysed with a Perkin Elmer EnVision 2103.

**Photography and wide-field microscopy.** Photographs were taken with a Nikon SLR camera under normal light and ultraviolet (365 nm) from a Spectroline ultraviolet lamp in a viewing cabinet. Wide-field optical and fluorescence images were taken with a Nikon SMZ1500 with DS-Fi1 camera peripheral and NIS Elements BR 3.2 64-bit image controller software.

**Liquid chromatography/mass spectrometry.** LC/MS was conducted on a Bruker HCT-Ultra, with data analysis undertaken using Compass Open Access in-built software. Samples were prepared in water and analysed in high-mass mode against a positively charged column.

**Fluorescence-lifetime imaging microscopy.** FLIM in the time-correlated single-photon counting (TCSPC) mode allows the fluorescence lifetime of species to be determined. Relaxation of a population of excited fluorescent molecules follows an exponential decay. By counting the number of photons arriving at a detector after an excitation pulse at picosecond resolution, a decay curve can be plotted and the lifetime value extracted. The lifetime value is expected to be dependent on the immediate environment of fluorophores. When TCSPC is combined with a scanning microscope set-up, an image with fluorescence lifetimes as pixel values is obtained, which is invaluable for interpreting the effect of dye environment within a crystal.

Single-photon fluorescence lifetime imaging was performed at the OCTOPUS imaging cluster at the Central Laser Facility located at the Research Complex at Harwell, UK. Briefly, an acousto-optical beam splitter was used to select 488 nm excitation light from the output of a picosecond pulsed supercontinuum light source operating at 78 MHz repetition rate (SuperK Extreme, NKT). Optical transmission and confocal fluorescence images of calcite doped with GREEN were acquired with a Leica TCS SP8 confocal microscope using a 60 × 1.4 numerical aperture oil immersion objective. For lifetime imaging of the same field of view, the fluorescence emission was detected with hybrid detectors and a TCSPC module (PicoHarp 300, PicoQuant) recorded the photon arrival times at each pixel of the image. Acquisition and analysis of lifetime images was handled by SymPhoTime software (PicoQuant). For regions of interest, a global fluorescence decay was obtained by combining the time-tagged photons from these pixels. A single exponential decay model was fit to each global decay, yielding a single-fluorescence lifetime. Data analysis was undertaken on Fiji (ImageJ).

**Atomic force microscopy.** AFM was conducted using a Bruker Multimode 8 with a Nanoscope V controller. The experiments were performed in both contact mode and tapping mode using silicon nitride cantilevers (model SNL-10, Bruker). Seeded growth experiments were performed using crystals grown on a glass slide cleaned with piranha solution. Seed crystals were grown by the ammonia diffusion method in [$Ca^{2+}$] = 10 mM. Growth solutions were prepared using stock solutions of $CaCl_2$, NaCl, GREEN, $NaHCO_3$ and $Na_2CO_3$. The growth solutions used to produce the data presented in Supplementary Fig. 2 had a [$NaHCO_3$] = 4.8 mM, [NaCl] = 85.8 mM and [$CaCl_2$] = 2.4 mM. [GREEN] = 0, 0.01 and 0.1 mM were investigated in these preliminary AFM experiments.

**High-resolution powder X-ray diffraction.** This procedure was the same as that used in a previous publication[36]. High-resolution powder X-ray diffraction measurements were carried out at the dedicated high-resolution powder diffraction beamline (I11) at the Diamond Synchrotron Radiation Facility (Diamond Light Source Ltd, Didcot, UK). The beamline is equipped with a crystal monochromator (a liquid nitrogen-cooled double-crystal silicon monochromator) and a crystal analyser at the incident and diffracted beams, respectively. The optics of the diffracted beam consists of nine (111) Si crystal analysers and the use of the advanced analysing optics yielded diffraction spectra of superior quality that exhibited intense and extremely narrow diffraction peaks with an instrumental contribution to the peak widths not exceeding 0.004°. Instrument calibration and wavelength refinement were performed with silicon standard samples from the National Bureau of Standards and Technology (NIST; Gaithersburg, MD, USA). Powders for analysis were loaded into 0.7 mm borosilicate glass capillaries, and to avoid intensity spikes from individual crystallites, the samples were rotated during measurements at a rate of 60 r.p.s. using high-resolution multi-analyser crystal diffraction scans, with scan times of 1,800 s. Spectra were recorded at room temperature.

**Normal-resolution powder X-ray diffraction.** Normal-resolution powder X-ray diffraction measurements were carried out on Phillips X'Pert MPD with Cukα ($\lambda = 0.15418$ nm) radiation. Scans were taken across different ranges depending on the sample as a scan rate of 2° per min with 10 mm slits on a spinning Si wafer sample holder.

**Crystal morphology modelling.** Modelling of crystal morphologies for generating graphical descriptions of approximate faces was conducted on WinXMorph.

**Data availability.** All data that support the findings of this study are available in the Research Data Leeds Repository 'David C. Green (2016): Dataset for 3D Visualisation of Additive Occlusion and Tunable Full-Spectrum Fluorescence in Calcite. University of Leeds. [Dataset].' with the identifier http://doi.org/10.5518/97

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

## Acknowledgements

This project was supported by an Engineering and Physical Sciences Research Council (EPSRC) Leadership Fellowship (FCM and JI, EP/H005374/1), and EPSRC grant EP/I001514/1] (D.C.G. and F.C.M.), where this programme grant funds the Materials in Biology (MIB) consortium. EPSRC grants EP/J018589/1 (Materials World Network, F.C.M. and Y.Y.K.) and EP/K006304/1 (F.C.M. and A.N.K.) are also acknowledged. B.M. and F.C.M. acknowledge the funding received from the European Commission under the FP7 project SMILEY (grant agreement no. 310637). We also thank the Central Laser Facility for time under proposal 15130010 and Diamond Light Source for time on beamline I11 under proposal EE10137. We thank Werner Kaminsky for the development and open access of WinXMorph. Finally, D.C.G. dedicates this work, with heartfelt joy, to Erik Roth Green, who arrived 5 January 2016.

## Author contributions

F.C.M. ran the project; D.C.G. designed and conducted crystal growth experiments; J.I., C.L., C.J.T., S.E.D.W. and D.C.G. performed sample preparation and analysis by FLIM; B.M., M.A.L., C.T. and D.C.G. performed sample preparation and data acquisition by high-resolution powder X-ray diffraction using synchrotron radiation; Y.-Y.K. performed data analysis of high-resolution powder X-ray diffraction data; P.D.T. and D.C.G. designed and performed synthetic experiments for RED; M.A.H. performed preliminary AFM studies; A.N.K. assisted with photography and figure design; F.C.M. and D.C.G. wrote the paper with contributions from all authors.

## Additional information

**Competing financial interests:** The authors declare no competing financial interests.

