## [Peer Review File · Nature Communications]

Reviewers' comments:

Reviewer #1 (Remarks to the Author):

General Comments:

This manuscript presents some interesting data on the variations in occlusion of organic additives within calcite crystals, and as suggested by the authors, results from varying kinetics of the obtuse and acute growth steps modified by adsorption and desorption kinetics, the latter of which are dependent on supersaturation. Although they find evidence of sectorization, other groups (Marc McKee, Graeme Hunter, etc.) have shown similar and even nicer examples of sectorization in calcium oxalate (which is a rather similar crystal), which should have been cited here. The older literature tends to discuss adsorption of additive to crystal faces, rather than hillock steps; but the sectorization concepts are still related. On the other hand, the FLIM study was a nice extension of these types of studies. The authors then show that different fluorophores can be incorporated, and even simultaneously, which then leads to an interesting cascade of energy transfer to yield white fluorescence. They point out that their tunable fluorescence might be of commercial interest for biodegradable and biocompatible calcite nanoparticles.

Other than the lack of full literature review, the only other weakness of the paper was the lack of an explanation of why the ACC was less effective at incorporating the fluorophores, which seems counterintuitive; so if there is any experiment that can help to ascertain the reason, that would be nice. Other than that, I thought the methodology was well designed, and the data was convincing (although some indication of number of samples that were imaged and showed similar features should be provided). The rest of the discussion and conclusions seem well thought out and valid, although I think the fluorescence evidence is more indirect than direct, as the wording in the discussion seems to imply.

Overall, the paper is written well, but the figures have many errors in labeling, as pointed out below, along with some other minor comments:

Specific Comments:

- pg 4, line 27 etc.: Figure 1 labels are confusing in the text. Should Fig. 1a be Figure 1-iiiia? Are the vertical a,b,c supposed to represent the same crystals shown in the SEM a,b,c; or only correlate to the confocal crystals?
- Figure 1v caption: There is no a and b and c. Do you mean 1, 2 and 3?
- page 4: It would be helpful to more clearly define the difference between intra- versus intersectoral zoning. (and is inter the same as sectoral?)
- pg, 4, line 104: It's not clear to me why this confocal image represents intrasectoral zoning. There should be fluorescence in a section or subset of the sector. I don't intrasectoral refers to missing sectors, does it? And I would like to know how many samples were analyzed (I don't find that in the methods), since orientation of a crystal on a substrate can influence addition interactions quite strongly.
- pg 5, line 148: You refer to Figure 5bi, but the text describes the top image.
- pg. 5, line 154 & 155: Figure 5c and 5d do not exist, and should be 5iii etc.
- pg. 5, line 136: The explanation for the spectral shifts is based on "stacking of the aromatic rings in the dyes", but given that they all have multiple sulfate groups, wouldn't there be electrostatic repulsion in these dyes? Or is there some ion bridge that favors this stacking?
- pg. 8, line 244: "revealed near-cubic 55 nm calcite particles", do you mean rhombic? (or are you just indicating they are slightly faceted?)
- pg 8, Discussion first sentence: "Despite the huge amount of literature concerning CaCO₃

precipitation in the presence of organic additives, little is known about whether these additives occlude within the crystal lattice." There are many papers showing occlusion of additives. Estroff in particular on strength of binding interactions; and all the CaOx work, such as those cited below.

- pg. 11, line 344: "To-date, information about the location of organic additives within calcite has come from single crystal XRD studies of biomaterials and synthetic crystals precipitated in the presence of additives." Actually, there are cathodoluminescence studies on calcite with various zoning patterns that are likely to be relevant here. (Carbonate Microfabrics; - Part of the series *Frontiers in Sedimentary Geology* pp 243-252; *Compositional Zoning and Crystal Growth Mechanisms in Carbonates: A New Look at Microfabrics Imaged by Cathodoluminescence Microscopy*, by Jeanne Paquette, W. Bruce Ward, Richard J. Reeder)
- pg. 11, line 366: "it is noted that dye occlusion in calcite was more efficient than in ACC". This is interesting and not necessarily what one might expect, so some discussion about this finding would be nice.

Papers that seem relevant and should probably be cited:

- Chien, Yung-Ching, David L. Masica, Jeffrey J. Gray, Sarah Nguyen, Hojatollah Vali, and Marc D. McKee. "Modulation of Calcium Oxalate Dihydrate Growth by Selective Crystal-Face Binding of Phosphorylated Osteopontin and Polyaspartate Peptide Showing Occlusion by Sectoral (Compositional) Zoning." *THE JOURNAL OF BIOLOGICAL CHEMISTRY* 284:35, no. August 28 (2009): 23491-501.
- Hunter, G. K., B. Grohe, S. O'young Jeffrey, J., E.S. Sørensen, and H. A. Goldberg. "Role of Phosphate Groups in Inhibition of Calcium Oxalate Crystal Growth by Osteopontin." *Cells Tissues Organs* 189(1-4) (2008): 44-50.
<http://dx.doi.org/http://search.proquest.com/docview/222627111?accountid=10920>
- Bullard, Theresa, John Freudenthal, Serine Avagyan, and Bart Kahr. "Test of Cairns-Smith's 'Crystals-as-Genes' Hypothesis." *Faraday Discussions* 136, no. 0 (2007): 231-45.
<http://dx.doi.org/10.1039/B616612C>.
- Incorporation of fluorescent molecules and proteins into calcium oxalate monohydrate single - Scientific Figure on ResearchGate. Available from:
https://www.researchgate.net/223505142_fig1_Fig-2-Intersectoral-zoning-in-calcium-oxalate-monohydrate-crystallized-in-the-presence [accessed 24 Jun, 2016]
- Face-specific binding of prothrombin fragment 1 and human serum albumin to inorganic and urinary calcium oxalate monohydrate crystals
Alison F. Cook, Phulwinder K. Grover, and Rosemary L. Ryall
BJU Int. 2009 Mar; 103(6): 826-835.

Reviewer #2 (Remarks to the Author):

I like this paper. It is terrific work that builds on data from the recent and remote pasts, but with a keen eye for technologically relevant questions such as white light emitters. Calcite LEDs, why not? The paper is lavishly illustrated, one of its great strengths. I encourage publication though there are some comments below that I hope the authors will consider.

There are two important citations that are missing that would enrich the discussion. The first is reference to the first calcite dye inclusion Kohlschütter, V.; Egg, C. *Über Wirkungen von Farbstoffzusätzen auf die Krystallisation des Calciumcarbonats.* *Helv. Chim. Acta* 1925, 8, 697–703) that we highlight in the table in our review on dyeing crystals (B. Kahr, R. W. Gurney, *Dyeing crystals*, *Chem. Rev.* 2001, 101, 893–951). This observation was recently repeated by Momper et al., *Langmuir*, 2015, 31, 7283 who showed by AFM that Congo red appeared to stabilize calcite surfaces and suppress etch pits.

This is the first paper of dyed crystals to employ fluorescence lifetime imaging as an additional contrast mechanism. While new contrast mechanisms are valuable because they invariably illustrate new things, the lifetimes varying from 1 - 5 ns, are virtually uninterpretable. So, these numbers, distinct as they may be, die on the vine. In this context, I point out two general considerations that are lacking in my opinion and that would strengthen the paper.

1. There is no consideration of polarization of the excitation or emission. This is strange. This is how structure is best determined in crystals of this kind. Polarization is far more informative than excited state lifetimes.

2. There is then no consideration of specific host / guest interactions on surfaces that justify incorporation here rather than there (and that would support polarization data).

One could make the argument that there is little room for additional information in this communication. And the white emission is undeniable. Therefore, deferring these issues to future publications where the mechanism is attacked in greater depths seems appropriate to me.

Additionally:

GREEN (pyranine) is a well known optical acid base indicator. It emits green in neutral solution and blue in acid. Dual fluorescence is frequently observed that corresponds to emission from the protonated and deprotonated excited states. While aggregation may be the reason for variance in fluorescence, there are other possibilities that ought to be considered.

On the false color scales in Figures 1 and 3, there should be a unit even though it is obvious the red means "hot", high intensity of luminescence.

Reviewer #3 (Remarks to the Author):

The manuscript successfully shows how to manipulate the incorporation of organic additives into calcite / release of the additives from calcite by changing the initial supersaturation with respect to the final calcitic polymorph. As stated in the abstract it is the first time that I encounter this specific zoning for this system and as such it is novel and important.

One point of criticism is the somewhat descriptive nature of the work; the data is beautiful, the discussion on the incorporation mechanisms of the different dyes stays a bit too much on the superficial side.

- 1) In the light of recent developments on calcium carbonate formation (nucleation vs. liquid phase separation) one could go deeper into differences with respect to calcite formation mechanisms. Although calcite mostly forms via a dissolution-precipitation mechanism, evidence for a particle-attachment mechanism of ACC particles attaching to an existing nucleus can be found in literature. Additionally, ACC particles formed at different supersaturations might behave differently, i.e. at high concentrations smaller particles are formed that will dissolve more readily than the ones formed at lower supersaturations. These mechanisms might reflect themselves in the distribution of the green label (or not), and a short discussion on this point might be appropriate. Secondly, at supersaturations < 5 mM no ACC precursor is formed but likely vaterite, that then transforms into calcite. Though vaterite is known not to take up any impurities, could the vaterite precursor have any role in the distribution of the label? It might be expected that the first calcite nucleus forms at the interfaces of existing calcium carbonates (that are low in fluorescent label), and in such a way could explain low amounts of GREEN in the center of most crystals.

2) In general; incorporation of the dye into calcite is related to the electrostatic interaction between the dye and the growing interface, kink site, step edge etc.. (this is what the trend in dye inclusion also shows, i.e. mostly related to the amount of sulfonate groups rather than molecular weight). This result therefore doesn't exclude the fact that there might still be specific electrostatic interactions between some of the planes and the dye, as some planes might have an overall lower or higher net charge than others, and it is not only the amount and availability of step edges. However, this might not be expressed in the system presented due to the conditions chosen. Changes in ionic strength, pH, Ca/Co₃ ratio in the solution will all heavily affect this interaction and might diminish or enhance such an effect. A more detailed discussion on this point would be appropriate, rather than stating the complexity of the system.

Response to Reviewers of NCOMMS-16-12303

We would like to thank the reviewers for their thorough and careful assessments of our paper “3D Visualisation of Additive Occlusion and Tunable Full-Spectrum Fluorescence in Calcite”, where these have undoubtedly enabled us to improve the quality of our paper.

Reviewer 1:

1. *Although they find evidence of sectorization, other groups (McKee, Hunter, etc.) have shown similar and even nicer examples of sectorization in calcium oxalate (which is a rather similar crystal), which should have been cited here. The older literature tends to discuss adsorption of additive to crystal faces, rather than hillock steps; but the sectorization concepts are still related.*

We actually were aware of the work referred to (and agree that calcium oxalate offers some nice examples of zoning), but had made a conscious decision to focus our paper on CaCO₃ alone – where this is reflected in the title of the paper. There are in fact many good examples of zoning in crystals, and we simply didn't have space to discuss all of these - we are already right at the word limit for *Nat Commun*. There also didn't seem to be any particular reason to single out calcium oxalate.

However, there are some important differences between the growth mechanisms of CaCO₃ and Ca oxalate, which make the example of Ca oxalate worth highlighting. We have therefore added the text given below, which we believe adds to the value of the discussion. We should also point out that *Nat Commun* allows a maximum of 70 references, so we have not been able to add all of the references suggested (so have selected the best ones) but have removed a couple of references from the introduction which we felt were less important than those for enhancing the discussion.

P2. “...sorting²; catalysis³; lighting⁴; and delivery.⁵ Organic macromolecules offering cavities, such as cyclodextrins, have also been widely explored for use as biosensors and drug delivery agents.⁶” (4 references removed from this section).

P12. “Adding to this discussion, it is also valuable to compare the mechanisms that give rise to zoning in calcite with those that operate for another common biomineral, calcium oxalate, which readily exhibits zoning effects.^{64, 65, 66} Unlike calcite – which only displays smooth {104} faces – calcium oxalate monohydrate and calcium oxalate dihydrate can exhibit smooth faces corresponding to a number of different crystal planes. Significant differences in additive binding energies to different crystal faces have been reported, which will facilitate inter-sectorial zoning.^{67, 68} A key feature of additive binding to calcite is the selectivity between the acute and obtuse steps, where this drives intra-sectorial zoning. Studies of peptide-binding to the different step edges present on given calcium oxalate faces suggest that a much smaller difference in binding energy may exist,^{67, 68} as compared with that estimated for aspartic acid binding to the acute and obtuse steps on calcite.⁴⁷ This makes the observation of intra-sectorial zoning less likely for calcium oxalate as compared with calcium carbonate.”

2. *Other than the lack of full literature review, the only other weakness of the paper was the lack of an explanation of why the ACC was less effective at incorporating the fluorophores, which seems counterintuitive; so if there is any experiment that can help to ascertain the reason, that would be nice.*

This is indeed an interesting question, and the referee is quite correct that we did not discuss this in our manuscript. We cannot think of any experiment we could readily perform that would give us a definitive answer to this (one would perhaps require modelling studies?), but it is certainly appropriate that we address this in our text. We have therefore added the following discussion to the section on ACC, where we feel it fits the best.

P6. "That the ACC contained significantly less GREEN than calcite crystals grown in the same concentration of dye is also interesting and perhaps at first sight counter-intuitive, given that one may expect it to be easier to occlude dye into the disorganised ACC structure than the crystalline calcite phase. However, dye occlusion is a kinetically-driven process such that the mechanism by which the dye becomes occluded must be considered as well as the stability of the product. Our data therefore demonstrate that binding of the dye to the calcite steps, and subsequent overgrowth is a more efficient process than coprecipitation with calcium and carbonate ions during the formation of ACC. "

3. *Overall, the paper is written well, but the figures have many errors in labeling, as pointed out below, along with some other minor comments:*

- pg 4, line 27 etc.: *Figure 1 labels are confusing in the text. Should Fig. 1a be Figure 1-iiia? Are the vertical a,b,c supposed to represent the same crystals shown in the SEM a,b,c; or only correlate to the confocal crystals?*

- *Figure 1v caption: There is no a and b and c. Do you mean 1, 2 and 3?*

- *pg 5, line 148: You refer to Figure 5bi, but the text describes the top image.*

- *pg. 5, line 154 & 155: Figure 5c and 5d do not exist, and should be 5iii etc.*

The reviewer has done a great job in identifying these errors. All have been corrected, as highlighted in the paper.

4. *It would be helpful to more clearly define the difference between intra- versus intersectoral zoning. (and is inter the same as sectoral?)*

Our definitions of inter-sectoral and intra-sectoral zoning were taken from the literature ("Dyeing Crystals" by Kahr and Gurney, *Chem. Revs.* 2001). However, on re-reading our text we note that these definitions were not put at the same place, where this undoubtedly makes it harder for the reader to see the difference between the two. We have also checked our text again and ensured that we always use the terms inter-sectoral and intra-sectoral, and do not abbreviate.

P4. "This is characteristic of intra-sectoral zoning, in which only a sub-volume of a single growth sector (associated with symmetry-related crystal faces) takes up the additive;18, 24 this can be attributed to preferential association of an impurity with specific hillock steps.25 Inter-sectoral zoning, in contrast, describes the uniform occlusion of dye within a particular growth sector"

P4. "...where such occupancy of symmetry-related sectors is indicative of inter-sectoral zoning. 24"

5. *pg, 4, line 104: It's not clear to me why this confocal image represents intrasectoral zoning. There should be fluorescence in a section or subset of the sector. I don't intrasectoral refers to missing sectors, does it? And I would like to know how many samples were analyzed (I don't find that in the methods), since orientation of a crystal on a substrate can influence addition interactions quite strongly.*

The zoning given in this figure (Fig 1biii) is declared 'intra-sectoral' because dye occlusion does not occur in all symmetry-related growth sectors. Dye is missing in obtuse-directed growth sectors due to the weaker interaction between the dye and obtuse steps. We can't rationalise this distribution based on different crystallographic faces. This is consistent with the definition of intra-sectoral zoning provided in "Painting Crystals" by Kahr and Vasquez.

Multiple samples (at least 3) were analysed from each population of crystals, where lower magnification images of these bulk populations are shown in the supplementary information. Our data demonstrate that occlusion is not determined by the crystal orientation. This can be seen by

comparing Figures 1b and 2. These samples were grown under the same conditions but exhibit different orientations. Despite this, they show the same dye distribution in 3D.

This is an important point, so we have now directly commented on this in the paper:

P4. "It is also important to note that the pattern of dye occlusion seen in these crystals is not determined by their orientation on the substrate. The crystals shown in Figures 1b and 2 are precipitated under the same conditions but show different orientations. Never-the-less, they exhibit identical occlusion patterns."

P1 (SI). "Low magnification images were obtained to demonstrate dye distribution across a population of composites on the same microscope slide, and higher magnification z-stacks were performed on at least 3 individual crystals per sample."

6. - pg. 5, line 136: *The explanation for the spectral shifts is based on "stacking of the aromatic rings in the dyes", but given that they all have multiple sulfate groups, wouldn't there be electrostatic repulsion in these dyes? Or is there some ion bridge that favors this stacking?*

This is an interesting question. We suggested that the dyes stack as this provides the best explanation for our experimental observations. The phenomenon of stacking of charged dye molecules is well-recognised in the literature. Looking at the example of stacking of TSPP molecules (5,10,15,20-Tetrakis(4-sulfonatophenyl)porphyrin, which are comparable in structure to the HPTS used here) it is recognised that at low pH, the inner amine groups protonate to yield a 2+ charge on the interior, while the outer 4 sulfonate groups remain deprotonated, such that there is a net negative charge overall. Stacking is thus favoured by an electrostatic contribution between these cationic and anionic groups, and through the pi-pi stacking contribution.

For the pyranine system described here, it is reasonable that this is facilitated by both pi-stacking and a charge bridging mechanism. Confirmation of this could form the basis of a further dedicated study. We have added the following text to the paper to address this question:

P5. "As HPTS is anionic under the reaction conditions used, such stacking could be promoted by ion-bridging by Ca^{2+} ions, and the contribution of π - π stacking."

7. pg. 8, line 244: *"revealed near-cubic 55 nm calcite particles", do you mean rhombic? (or are you just indicating they are slightly faceted?)*

These nanoparticles simply appear 'near-cubic' in morphology – where exactly this expression has been used to describe them in the literature. To eliminate any confusion we have simply deleted this term.

8. - pg 8, Discussion first sentence: *"Despite the huge amount of literature concerning CaCO_3 precipitation in the presence of organic additives, little is known about whether these additives occlude within the crystal lattice." There are many papers showing occlusion of additives. Estroff in particular on strength of binding interactions; and all the CaOx work, such as those cited below.*

We were specifically talking about additive occlusion within calcite, not the effect of organic additives on calcite crystal nucleation and growth (where we agree there is an enormous amount of work). As this wasn't clear, we have re-written this section of the text. We do not have space to review the effect of organic additives on all crystal types – the volume of literature is vast.

P9. "There is a huge amount of literature on the influence of organic additives on the precipitation of CaCO₃, where these have focused on the effect on crystal polymorph and morphology.³⁹ Due to the challenge of quantifying the amounts of organic molecules present within a carbonate-based crystal, much less is known about whether association of the additives with the growing crystals is accompanied by their incorporation into the crystal lattice. It is well-recognised that proteins are occluded within calcite single crystal biominerals, where these are quantified by gravimetric analysis after dissolution of the mineral.⁴⁰ Our study provides new insight into the occlusion of small organic molecules (here fluorescent dyes) within calcite single crystals, and in particular into the existence of zoning and the relationship between occlusion and changes in crystal morphologies. It is noted that proteins tagged with fluorescent dyes have been occluded within polycrystalline and single crystal calcite, but zoning was not observed.⁴¹ The dye Congo Red has also been reported to occlude within calcite,⁴² and a recent AFM study has demonstrated that this dye stabilises the polar step edges along the [010] direction of calcite (10.4).⁴³"

9. - pg. 11, line 344: *"To-date, information about the location of organic additives within calcite has come from single crystal⁵⁹, 60, 61 and powder⁶² XRD studies of biominerals and synthetic crystals precipitated in the 346 presence of additives.:"* Actually, there are catholuminescence studies on calcite with various zoning patterns that are likely to be relevant here. (Carbonate Microfabrics; - Part of the series *Frontiers in Sedimentary Geology* pp 243-252; *Compositional Zoning and Crystal Growth Mechanisms in Carbonates: A New Look at Microfabrics Imaged by Cathodoluminescence Microscopy*, by Jeanne Paquette, W. Bruce Ward, Richard J. Reeder)

We actually had a discussion of this in an earlier (longer) version of our manuscript, but removed it to reduce the final word count. We totally agree with this comment and are delighted to reinstate this text.

P9: "That soluble additives can preferentially occlude within specific zones in calcite single crystals is well-recognised for ions such as Mg²⁺, Mn²⁺, Sr²⁺,⁴⁸, 49, 50 and SO₄²⁻,⁵¹ where detailed investigations by AFM have demonstrated preferential binding of the smaller ions (Mg²⁺ and Mn²⁺) to the acute step edges, and the larger ions (Sr²⁺ and SO₄²⁻) to the obtuse. Such preferential binding of additives to the distinct calcite steps results gives rise to zoning effects."

Reviewer 2:

1. *There are two important citations that are missing that would enrich the discussion. The first is reference to the first calcite dye inclusion Kohlschütter, V.; Egg, C. Über Wirkungen von Farbstoffzusätzen auf die Krystallisation des Calciumcarbonats. Helv. Chim. Acta 1925, 8, 697–703) that we highlight in the table in our review on dyeing crystals (B. Kahr, R. W. Gurney, Dyeing crystals, Chem. Rev. 2001, 101, 893–951). This observation was recently repeated by Momper et al., Langmuir, 2015, 31, 7283 who showed by AFM that Congo red appeared to stabilize calcite surfaces and suppress etch pits.*

These are excellent suggestions and have been added to the script as part of the discussion.

P9. "The dye Congo Red has also been reported to occlude within calcite,⁴² and a recent AFM study has demonstrated that this dye stabilises the polar step edges along the [010] direction of calcite (10.4).⁴³"

2. *This is the first paper of dyed crystals to employ fluorescence lifetime imaging as an additional contrast mechanism. While new contrast mechanisms are valuable because they invariably illustrate new things, the lifetimes varying from 1 - 5 ns, are virtually uninterpretable. So,*

these numbers, distinct as they may be, die on the vine. In this context, I point out two general considerations that are lacking in my opinion and that would strengthen the paper.

Although these values are indeed difficult to interpret from the graphs shown, we do not really agree that they have no meaning. Fitting of these curves for these samples reproducibly (multiple samples were measured) demonstrates that the values are dependent on the local environment of the dye, and that this environment changes depending on where the dye is distributed. Of course, with the evidence we have, we cannot give a conclusive statement about the origin of the lifetime changes, which may be the source of Reviewer 2's concern? Despite this, some reasonable hypotheses (hydration and stacking) are offered. The origin of these effects could possibly be determined with further studies.

3. *There is no consideration of polarization of the excitation or emission. This is strange. This is how structure is best determined in crystals of this kind. Polarization is far more informative than excited state lifetimes.*

The referee is completely correct that this would be really interesting to do. However, we simply don't have the capability within our group, or at Leeds Uni. We also contacted the UK Central Laser Facility in Didcot (where the majority of our advanced microscopic experiments were conducted), but they again don't yet have this capability (although they may acquire it next year, depending on funding). We have therefore been unable to make these measurements.

4. *There is then no consideration of specific host / guest interactions on surfaces that justify incorporation here rather than there (and that would support polarization data).*

We agree that we cannot make any statement of why the dyes have different affinities to the acute and obtuse step edges. The best way to answer this would be to do modelling studies, which is certainly something we can explore if we can persuade a modelling group to collaborate with us. This is definitely something worth looking at in the future.

5. *GREEN (pyranine) is a well-known optical acid base indicator. It emits green in neutral solution and blue in acid. Dual fluorescence is frequently observed that corresponds to emission from the protonated and deprotonated excited states. While aggregation may be the reason for variance in fluorescence, there are other possibilities that ought to be considered.*

While CaCO_3 has many attractive features as a model crystal system, one failing is that it can only be precipitated under basic conditions (experiments are typically performed at pH 8 and above). It is therefore not possible to investigate the effect of pH on the state of the occluded dye.

All of the measurements of the fluorescence spectra (Fig 4) were conducted at pH 10.6, where pyranine is fully deprotonated. It therefore seems rather unlikely that the spectral variations are due to variations in the protonation of the dye.

6. *On the false color scales in Figures 1 and 3, there should be a unit even though it is obvious the red means "hot", high intensity of luminescence.*

This has now been addressed in the Figure captions. "Colour scale: Blue (low intensity) to Red (high intensity); Black = no signal, White = detector saturation."

Reviewer 3:

1. *One point of criticism is the somewhat descriptive nature of the work; the data is beautiful, the discussion on the incorporation mechanisms of the different dyes stays a bit too much on the superficial side.*

Without further extensive AFM studies and perhaps modelling studies (which would form a separate publication in their own right), we do not really know how we can add further weight to our discussion. As it is, we did perform preliminary AFM experiments, which fully supported our suggested mechanism. Further, one can argue that occlusion of dye molecules itself provides a beautiful way of investigating the mechanism of additive interaction with calcite – where this simply depends on optical microscopy. However, we believe that it is wrong to write in a tone that suggests that we have provided conclusive proof of mechanism because this is not true.

2. *In the light of recent developments on calcium carbonate formation (nucleation vs. liquid phase separation) one could go deeper into differences with respect to calcite formation mechanisms. Although calcite mostly forms via a dissolution-reprecipitation mechanism, evidence for a particle-attachment mechanism of ACC particles attaching to an existing nucleus can be found in literature. Additionally, ACC particles formed at different supersaturations might behave differently, i.e. at high concentrations smaller particles are formed that will dissolve more readily than the ones formed at lower supersaturations. These mechanisms might reflect themselves in the distribution of the green label (or not), and a short discussion on this point might be appropriate. Secondly, at supersaturations < 5 mM no ACC precursor is formed but likely vaterite, that then transforms into calcite. Though vaterite is known not to take up any impurities, could the vaterite precursor have any role in the distribution of the label? It might be expected that the first calcite nucleus forms at the interfaces of existing calcium carbonates (that are low in fluorescent label), and in such a way could explain low amounts of GREEN in the center of most crystals.*

This is one comment that we do not agree with. We have not investigated how the dyes affect the nucleation/ early stages of growth of calcite crystals. Our data (which shows the location of dyes with calcite crystals over 10 μm in size) simply gives information on how the dyes interact with calcite crystals **during their growth phase**. Further, we ran experiments under conditions where ACC is not produced as a precursor, and observe similar rhombohedral calcite crystals and occlusion patterns. Vaterite was not also observed as a precursor phase.

As a side-note, we have also been conducting experiments on additive-directed growth of CaCO_3 using a crystal hotel microfluidic device (publication in preparation). Interestingly perfect calcite rhombohedra are seen at early stages of growth, and it is not until the crystals reach sizes of at least 100 nm is any effect of the additive seen on the crystal morphology.

3. *In general; incorporation of the dye into calcite is related to the electrostatic interaction between the dye and the growing interface, kink site, step edge etc.. (this is what the trend in dye inclusion also shows, i.e. mostly related to the amount of sulfonate groups rather than molecular weight). This result therefore doesn't exclude the fact that there might still be specific electrostatic interactions between some of the planes and the dye, as some planes might have an overall lower or higher net charge than others, and it is not only the amount and availability of step edges. However, this might not be expressed in the system presented due to the conditions chosen. Changes in ionic strength, pH, Ca/Co3 ratio in the solution will all heavily affect this interaction and might diminish or enhance such an effect. A more detailed discussion on this point would be appropriate, rather than stating the complexity of the system.*

It is correct that the electrostatic interaction between the exposed crystal face and the additive host/dye interaction will undoubtedly affect occlusion. However, **calcite only expresses smooth {104} faces**. When calcite crystals (grown in the presence of additives) exhibit rough faces that are parallel to non-{104} faces, these simply comprise a pile-up of {104} steps. Therefore, there are no

other faces for the dyes to adsorb to. The reaction conditions (supersaturation etc) do indeed affect the zoning patterns seen, as shown in our data. Thus, the dyes can be uniformly occluded, or only incorporated within specific zones, according to the growth conditions.

REVIEWERS' COMMENTS:

Reviewers #1 and #2 wrote comments to the editor only. Both find the revisions satisfactory.

Reviewer #3 (Remarks to the Author):

After the first revision an appropriate answering on the reviewers' comments, this reviewer doesn't have any reservations anymore.